



# Biologically effective daily radiant exposure for erythema appearance, previtamin D₃ synthesis and clearing of psoriatic lesions from erythema biometers at Belsk, Poland, for the period 1976-2023

Janusz W. Krzyścin[1], Agnieszka Czerwińska[1], Bonawentura Rajewska-Więch[1], Janusz Jarosławski[1], Piotr S. Sobolewski[1], Izabela Pawlak[1]

1Institute of Geophysics, Polish Academy of Sciences, Warsaw, 01-452, Poland

*Correspondence to*: Janusz W. Krzyścin (jkrzys@igf.edu.pl)

**Abstract.** A long-term series of exposures to solar ultraviolet (UV) radiation is required to assess the risks and benefits of radiation on different human biological processes. However, homogenisation of the amount of biologically effective solar energy reaching the Earth's surface over long periods (i.e. energy weighted according to the sensitivity of the selected biological process to solar radiation) is challenging due to changes in measurement methods and instruments. This paper presents the world's longest homogenised time series of biologically effective daily radiant exposures (DRE) from regular monitoring with different erythemal biometers (EB) operated at the Central Geophysical Laboratory of the Institute of Geophysics, Polish Academy of Sciences (IG PAS), Belsk (20.79°E, 51.84°N) from 1 January 1976 to 31 December 2023. The following biological effects were considered: the appearance of erythema, cutaneous synthesis of previtamin D₃, and clearing of psoriatic lesions. The data for the latter two biological effects are estimated based on the proposed method of using EB measurements to calculate other non-erythemal DRE. The following broadband erythemal radiometers were used in the monitoring: Robertson-Berger (1975−1992), Solar Light model 501 (1993−1994 with #927, 1995−2013 with #2011) and Kipp-Zonen UV-AE-T #30616 from 5 August 2013 to the present. From 1976 to 2013, the homogenisation procedure consisted of comparing the measured erythemal DRE and UV index (erythemal irradiance at noon) with the corresponding synthetic values from simulations using a radiation transfer model. Between 2014 and 2023, the raw data were compared with data from a collocated reference instrument, the Brewer Mark II #64 spectrometer. Such comparisons resulted in a set of multipliers that were applied to the raw EB measurements. Two different versions of the homogenisation method were applied (for erythemal DRE and UV index with different selection of cloudless days), and three regression models were constructed for the erythemal data based on total column ozone, aerosol optical depth and global irradiance clearness index. Linear trends calculated from reevaluated and reconstructed time series (a total of seven time series were considered) showed a statistically significant increase in erythemal annual and summer (June to August) radiant exposures of about 6 % per decade over the period 1976−2005. Thereafter, no trend was observed. The same trend estimates were found for all biological effects considered. The raw and reevaluated data are made freely available via the following repository: https://doi.org/10.1594/PANGAEA.972139 (Krzyścin et al., 2024). An additional version of the reevaluated data, together with the corresponding clear sky and proxy data used in the UV data reconstruction, is archived at https://doi.org/10.25171/InstGeoph_PAS_IGData_Biologically_Effective_Solar_Radiation_Belsk_1976_2023 (Krzyścin, 2024).



**Keyword(s)**: biometer; biologically effective irradiance, homogenisation, radiant exposure
**1 Introduction**
Molina and Rowland (1974), winners of the 1995 Nobel Prize in Chemistry, argued that man-made
chlorofluorocarbons (CFCs), which were widely used in industry in the 1970s, could penetrate the stratospheric
ozone layer where they were destroyed by short-wave ultraviolet (UV) radiation, releasing free chlorine atoms and
causing stratospheric $O_3$ depletion in the catalytic reaction cycle. Solar radiation in the shortest part of its spectrum
that reaches the Earth's surface (290−315 nm), known as UV-B, is strongly absorbed by stratospheric ozone. The
discovery of the ozone hole over Antarctica (Chubachi, 1984; Farman et al., 1985) and the predicted decreasing
trend in total column ozone ($TCO_3$) in other regions have stimulated interest in establishing continuous monitoring
of UV-B irradiance reaching the ground. In addition, there is growing evidence that such UV trends can cause
various adverse health effects, such as skin cancers (including the deadly melanoma), DNA damage,
immunosuppression, oxidative stress and skin ageing (Neale et al., 2023).
Solar UV-B radiation from space is attenuated as it passes through the atmosphere due to light scattering (by cloud
particles, atmospheric gases and aerosols) and absorption (by $O_3$, $NO_2$, $SO_2$ and aerosols). The attenuation of light
increases with its path length through the atmosphere (i.e. usually described by the air mass), so solar elevation
and ground surface altitude are key parameters to consider in surface UV modelling. Other factors forcing UV
variability at the surface that are often used as proxies for atmospheric UV-B attenuation are total column $O_3$
($TCO_3$) to account for UV absorption by ozone, the clearness index (CI) (i.e. a quotient of the all-sky global solar
irradiance (GSI) at the surface and the corresponding synthetic clear-sky value to account for combined
cloud/aerosol scattering effects on UV), and aerosol optical depth (AOD) in the UV (parameterising UV
attenuation by aerosols). $TCO_3$ and GSI have been found to be the most effective for modelling surface UV-B
radiation (Koepke et al., 2006, den Outer et al., 2010).
In the early 1970s, the broadband Robertson-Berger (RB) meter was developed to measure the biologically
effective (BE) UV radiation that causes skin redness, also known as erythema (Berger, 1976). The spectral
characteristics of RB resembled the erythemal sensitivity of human skin. RB instruments began continuous
monitoring of erythemal irradiance in 1974 at eight sites in the United States (Scotto et al., 1988). During the
1970s, instruments were operated in other countries (Austria, Australia, Germany, Poland, Sweden, Switzerland)
(WMO, 1977). At the beginning of this global network, RB meters were calibrated using a travelling standard
meter provided by the Photobiology Center at Philadelphia University. After a few years, at some stations,
including the Institute of Geophysics, Polish Academy of Sciences (IG PAS) station at Belsk (51.84°N, 20.78°E),
this calibration method was replaced by comparisons with values modelled by the radiative transfer model. The
Dave-Halpern model was used to estimate erythemally weighted irradiance for cloudless sky conditions to
calibrate the Belsk data (Słomka and Słomka, 1985). Serious drawbacks of RB measurements were their results in
relative units (counts), temperature sensitivity, a lot of manual work in data preparation, sometimes rapid ageing,
and difficulties in accurately converting counts into the so-called sunburn unit (the minimum erythemal radiation
exposure that causes redness of the skin). These problems were significantly reduced in a new version of the RB
meter, a prototype of the current UV biometer, developed in the late 1980s as a result of collaboration between IG
PAS and the Institute of Medical Physics of the University of Innsbruck (Blumthaler et al., 1989; Słomka and





Słomka, 1993). Further prototype work at Solar Light (SL) Co. in Philadelphia resulted in the production of a
commercial SL Biometer Mod 501A, which replaced the RB meter.
Other versions of broadband UV biometers for UV monitoring were introduced in the 1990s, including those from
Yankee Environmental Systems (Turner Falls, USA) and Kipp and Zonen (KZ) Co. (Delf, Netherlands). However,
there was a need to standardise the calibration procedure for the broadband UV meters as it became apparent that
the calibration provided by the manufacturer could not be relied upon even for the same type of instrument
(Leszczynski et al., 1998). A standard calibration method that takes into account the individual spectral
characteristics of the instrument and the loss of sensitivity has been proposed (Hülsen and Gröbner 2007).
However, uncertainties of ~7 % can still be expected for well-maintained biometers (Gröbner et al., 2009).
This article presents a retrospective evaluation of all UV measurements (1976−2023) at Belsk made with different
broadband instruments: RB (1976−1992), SL biometer model 501 A (SL501 A) (two instruments were used #927
and #2011 for the period 1993−1994 and 1995−2013, respectively) and KZ UV-AE-T #30616 (KZ616) from 5
August 2013 to the present. The reevaluation for the period 1976−2013 is based on a comparison of the
measurements with the synthetic daily erythemal irradiance and UV index (the midday value of erythemal
irradiance) from a radiative model simulation for clear sky conditions using $TCO_3$ and AOD measured at Belsk as
model input parameters. The quality of the KZ616 data (2013−2023) will be accessed through comparisons with
clear-sky erythemal irradiances simultaneously measured by the well-maintained Brewer spectrophotometer Mark
II #64 (BS64). Erythemal daily radiant exposures (DRE) for the entire period of the UV measurements at Belsk
will be transferred to the corresponding vitamin D3 and antipsoriatic DRE using a method proposed by Czerwińska
and Krzyścin (2024a). A comparison of these DRE with those from BS64 spectral measurements in the period
2014-2023 will indicate the accuracy of the proposed reconstruction method of past BE data based on a statistical
approach using typical proxies ($TCO_3$, GSI) characterising atmospheric UV attenuation. Finally, trend calculations
in annual (January−December) and summer (June−August) radiant exposures (RE) for all biological effects
considered and versions of the recalculated UV data from 1976−2023 will be presented to confirm the robustness
of the long-term changes in the BE radiation measured at Belsk.
**2 Materials and Method**
**2.1 UV monitoring**
The recording of solar erythemal irradiance with a standard RB meter (detector recorder #40), initiated in Belsk in
May 1975, was carried out until 1994. From May 1993, in parallel with the RB measurements, the monitoring of
erythemal irradiance using the SL Biometer 501 A #927 was initiated in order to establish monthly transfer
coefficients for converting the RB output in sunburn units (SU) into erythemal units, i.e. the minimum erythemal
dose (MED) causing skin redness in typical Caucasian skin, which was entered into the SL Biometer 501 A
measurements (Puchalski, 1995). It was assumed that MED=210 $J_{eryt}$ $m^{-2}$, where $J_{eryt}$ denotes spectral irradiance
integrated over time and wavelengths (290−400 nm) after weighting by the erythema action spectrum (CIE, 2019).
Simultaneous measurements continued until December 1994, and all erythemal DRE measured with the RB meter
before 1993 were multiplied by these transfer coefficients to obtain data comparable to those with the SL Biometer
501 A.
As the RB meter showed sensitivity to ambient temperature, a correction for temperature effect was applied to the
raw daily RB values (Borkowski, 1998) using empirical formulas proposed by Koskela et al. (1994). In addition,



the RB Belsk series was also found to be affected by a change in calibration method in 1985, as Dave-Halpern
model calculations for cloudless conditions replaced field comparisons with the travelling standard instrument
This resulted in a downward step change of 14 % in the UV series (Borkowski, 2000). The reevaluated time series
of erythemal DRE for the period 1976−1992 as made by Borkowski (2008) was archived and formed part of the
raw Belsk's erythemal time series (1976−2023), which is further homogenised in this study.
Subsequent UV measurements included SL501 A # 927 (1993−1994) and #2011 (1995−2013), which were only
pre-calibrated by the instrument manufacturer. In 2005, KZ616 was added to the IG PAS UV network and served
as the reference instrument. It was not used for everyday UV monitoring but only for occasional international
calibration campaigns to provide a source for further calibrations with our SL biometers operating in Belsk and
Hornsund (Spitzbergen). KZ616 started regular UV monitoring on 5 August 2013, replacing the previous
SL501 A #2011, as BS64 (normally measuring $TCO_3$ and Umkehr ozone at Belsk since 1992) was established as
the new UV reference instrument for the IG PAS network, which has been in operation until now. The performance
of KZ616 has proved to be very stable and is still involved in regular UV monitoring.

### 2.2 Ancillary data

Daily representatives of $TCO_3$ at Belsk are taken from the IG PAS data portal (Krzyścin, 2024), which contains
results of daily average $TCO_3$ measurements throughout the day, prepared for UV modelling purposes. For
example, the most reliable daily representative value of $TCO_3$ (marked with flag no. 1) was calculated as an average
of the most accurate measurements (the so-called direct sun measurements) made by the Dobson
spectrophotometer between 9:00 and 13:00 UTC. The least accurate case of ground-based $TCO_3$ observations
(with flag no. 5) occurred under cloudy and low sun elevation conditions, i.e. before 9:00 and after 13:00 UTC. In
this case, only the least reliable Dobson observations were available for calculating the daily $TCO_3$ representative
under overcast zenith and high air masses. In the rare cases when ground observations were not available, satellite
data (flag 6 or 7 depending on the data source) and/or $TCO_3$ reanalysis data (flag 8) were used.
CI is a commonly used measure of cloud attenuation of global (direct and diffuse) solar irradiance at ground level
(Liu and Jordan, 1960). Daily values of CI are calculated as the quotient of the all-sky (G) and the corresponding
synthetic clear-sky ($G_0$) daily integral of global solar irradiance. Typically, G is derived from observations and $G_0$
from a model simulation, depending on the amount of solar absorbers (mostly water vapour) and AOD. Global
solar DRE were obtained from routine monitoring of solar irradiance by various pyranometers (since 1965)
including the following instruments: Kipp CM 6, Sonntag PRM-2, Kipp & Zonen CM 11, and
Kipp & Zonen CM 21. The data were calibrated using the Polish national standard, which was previously
calibrated at the World Radiation Centre in Davos. In addition, the Campbell-Stokes sunshine recorder provided
the duration of sunshine per day to pre-select sunny days. All these data are archived in the IG PAS Data Portal
(Krzyścin, 2024).
To support the quality of the UV observations at Belsk, the long-term variability of BE radiance was also obtained
from the UV reconstruction models (Section 2.3) using proxies ($TCO_3$ and CI) from the ground-based observations
and reanalysis datasets. The European Centre for Medium-Range Weather Forecasts (ECMWF) v5 (ERA5)
reanalysis provides, in addition to many other variables, intra-day $TCO_3$ values, global solar irradiance for clear
sky and all−sky conditions for the period 1940−2024, which are freely available on the ERA5 (2024) website.
Also included are data (from 1 January 1980 to the present) downloaded from the Modern-Era Retrospective



Analysis for Research and Applications version 2 (MERRA-2) database (GMAO, 2024) using the Giovanni data
search tool, which is freely available on the Giovanni (2024) website.
Atmospheric aerosols can be significant drivers of surface UV radiation, especially under clear sky conditions
(Krzyścin and Puchalski, 1998). The column properties of aerosols can be obtained from ground-based
observations and used in the modelling of radiative transfer in the atmosphere. Aerosol properties are described
by various characteristics (e.g. including AOD, single scattering albedo, asymmetry factor). In this article, we use
Belsk's AOD at 340 nm (IG PAS Data Portal, Krzyścin (2024)), which is estimated from the Linke turbidity factor
measurements with Sonntag pyrheliometers between 1976 and 2013 (Posyniak et al., 2016) and from the co-
located solar photometer CIMEL CE 318-T (2014−2023) operating within the Aerosol Robotic Network
(AERONET) (AERONET, 2024). Other aerosol properties are kept constant and equal to their typical values for
the rural site.
**2.3 UV models**
**2.3.1 Clear-sky model**
Synthetic clear-sky values of BE (erythema appearance, previtamin $D_3$ synthesis, clearing of psoriasis lesions) RE
and irradiance at noon in day D, $RE_{EFF,CS}(D)$ in $J_{eff}$ m$^{-2}$, and $Ir_{EFF,CS}(t=$noon) in $W_{eff}$ m$^{-2}$, respectively, are derived
from look-up tables obtained from the Tropospheric Ultraviolet and Visible (TUV) radiation transfer model (TUV,

169 2024):

$$RE_{EFF,CS}(D) = \int_{Sunrise(D)}^{Sunset(D)} Ir_{EFF,CS}(t)dt \qquad (1)$$
$$Ir_{EFF,CS}(t) = \int_{290\,nm}^{400\,nm} Ir_{CS}(\lambda,t)\, AS_{EFF}(\lambda)d\lambda \qquad (2)$$
where $Ir(\lambda,t)$ is the spectral irradiance in time $t$ for the wavelength $\lambda$ and $AS_{EFF}(\lambda)$ denotes the action spectrum
for specific biological effect EFF: EFF=ERYT for erythema (CIE 2019), EFF=VITD3 for photosynthesis of
previtamin $D_3$ in human skin (CIE 2006), and EFF=PSOR for psoriasis clearing (Krzyścin et al., 2012). Figure 1

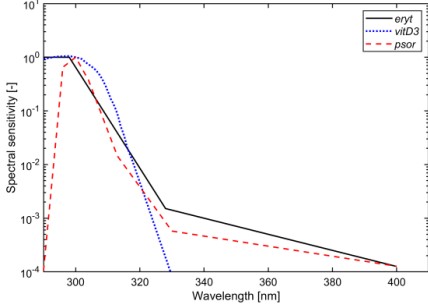

presents the action spectra used.

**Figure 1. Normalised action spectra for the specific biological effects: erythema appearance, photosynthesis of**
**previtamin $D_3$ in human skin, psoriasis clearing.**
Input to the clear-sky version of TUV model (daily representatives of $TCO_3$, annual and monthly mean AOD at
340 nm for the period 1976−2013 and 2014−2023, respectively) and output ($RE_{EFF,Clear-Sky}(D)$ and
$Ir_{EFF,Clear-Sky}(t$=noon)$, where EFF={ERYT, VITD3, PSOR}), are archived in IG PAS Data Portal (Krzyścin,
182 2024).

### 2.3.2 Reevaluation of the UV measurements

The intraday UV measurements at Belsk from 1976 to 2023 can be clearly divided into three periods:
1 January 1976−31 December 1992, 1 January 1993−4 August 2013, and 5 August 2013−31 December 2023,
according to the different broadband instruments used for UV monitoring, i.e. RB, SL501 A, and KZ616,
respectively. For the first period, only the daily erythemal RE was archived, whereas for other periods the
erythemal irradiances at noon, the so-called UV index (UVI), were also available. There were also periods when
both instruments were operated simultaneously for calibration purposes: March 1992−December 1994 (RB versus
SL501 A), 5 August 2013−31 December 2014 (SL501 A #2011 versus KZ616), and 5 August 2013−31 December
2023 (KZ616 versus BS64).
The calibration procedure before 5 August 2013 consisted of comparing the raw erythemal data with the
corresponding synthetic values obtained from the radiative model simulations (described in Sections 2.3.1) for the
days when clear sky conditions can be assumed from the ancillary data. The locally weighted scatterplot smoother
(LOWESS, Cleveland, 1979) was used to extract the smoothed pattern of the multipliers of the raw UV data, i.e.
the calibration coefficients (CCs), from the daily ratios between synthetic and erythemal REs (for version CC1 of
the calibration) or from the ratios between UVIs (version CC2) taken for the days when clear sky conditions can
be assumed at Belsk. Two sets of CCs were examined to determine the range of uncertainty in the CC estimates.
In order to allow for greater variability in the CC values, different criteria for clear sky conditions were applied,
and the smoothing procedure was applied to the long (1976−2013) and short (1993−2013) UV time series for the
CC1 and CC2 versions, respectively. Accordingly, the following conditions were applied for the selection of clear
sky sets:
• CC1 – direct sun $TCO_3$ measurements occurred between 9:00−13:00 UTC (code 1 for the $TCO_3$ observation
in IG PAS Data Portal, Krzyścin (2024) ) and the difference between observed sunshine duration and
theoretical one (for SZA < 85°) is less than 0.5 hour as for higher SZA broad band UV measurements are
unreliable and the Campbell-Stokes instruments starts when direct sun irradiance exceeded 120 W m$^{-2}$.
• CC2 – For $TCO_3$, the same condition was set as for CC1, and the ratio between the observed and theoretical
sunshine hours (for SZA < 85°) is not less than 85 %. CC2 values have only been calculated for the period
since 1 January 1993. Prior to this date, we assumed that the calibration coefficients were equal to 1.0
according to the recalibration of the RB data in 2011 (Krzyścin et al., 2011).
The CC1 and CC2 versions of the reevaluated Belsk UV data are stored in the following free-access data archives:
https://doi.org/10.1594/PANGAEA.972139 (Krzyścin et al., 2024) and
https://doi.org/10.25171/InstGeoph_PAS_IGData_Biologically_Effective_Solar_Radiation_Belsk_1976_2023
(Krzyścin, 2024), respectively.

### 2.3.3 Reconstruction of BE radiation from the erythemal data

Broad-band instruments for measurement of the erythemal irradiance can also estimate non-erythemal irradiance
by multiplying the erythemal irradiance by the so-called conversion factors ($CF_{EFF}$) derived from spectral UV
measurements and/or radiative transfer simulations (Schmalwieser et al., 2022; Czerwińska and Krzyścin, 2024a):



$Ir_{EFF}(t) = CF_{EFF}(TCO_3, SZA) \times Ir_{ERYT}(t),$                                   (3)
where $SZA$ denotes the solar zenith angle at time $t$. Following this concept, the daily radiant exposure for
previtamin D$_3$ synthesis and psoriasis clearance on the current $D$ day, $RE_{VITD3}(D)$ and $RE_{PSOR}(D)$, were estimated
using the daily conversion factor, $CF_{EFF}^*$, applied to the reevaluated erythemal DRE:
$RE_{EFF}(D) = CF_{EFF}^*(TCO_3, D^*) \times RE_{ERYT}(D),\ EFF = \{VITD3, PSOR\},$                  (4)
where $CF_{EFF}^*$ depends on $TCO_3$ and $D^*$ day of the year (i.e. between 1 and 365/366) corresponding to the current
$D$ day. $CF_{EFF}^*$ and $CF_{EFF}$ values were obtained from the radiative model simulations. The time series (1976−2023)
of these values and $RE_{EFF}(D)$ and $Ir_{EFF}(t = \text{noon})$ from Eq. (3–4) have been archived in the IG PAS Data Portal
(Krzyścin, 2024).

### 2.3.4 Regression models

Various regression models built from the UV data collected in the period 2014−2023 allowed for extended daily
erythemal RE analysis for the entire 1976-2023 period to provide a quality measure of the reevaluated UV data.
According to a frequently used UV modelling concept (e.g. Rieder et al., 2008; Outer et al., 2010; Čížková et al.,
2018; Czerwińska and Krzyścin, 2024b) that the erythemal DRE on the current day $D$, $RE_{ERYT}(D)$, is the product
of the so-called cloud modification factor ($CMF$), which is an empirical function of $CI$, and the synthetic clear-sky
value, $RE_{ERYT,CS}(D)$ (Section. 2.3.1):
$RE_{ERYT}(D) = CMF(CI(D)) \times RE_{ERYT,CS}(D),$                                  (5)
$CMF(CI(D))$ is parameterised as a power function with the regression coefficients, $\alpha$ and $\beta$, depending on SZA at
noon, $SZA_N$, for the current day $D$:
$CMF(CI(D)) = \alpha[CI(D)]^\beta,$                                                (6)
where estimates for the regression coefficients were obtained from the 2014−2023 data when the KZ616
measurements were well−fitted to the BS64 data (Section 3.1). In $CI$ calculations ($CI = GG_0^{-1}$), the global solar
DRE, G, comes from observations at Belsk or ERA5, and its clear-sky equivalent, $G_0$, from ERA5 (before 1980),
and thereafter the mean of ERA5 and MERRA-2 values.
The standard least-squares subroutine (Matlab function – $fitlm(x,y)$) provided the estimates for three arbitrarily
selected SZA ranges (Table 1). These regression coefficients were used for the reconstruction of the RE$_{ERYT}$(D)
time series for the entire period of UV measurements (1 January 1976 up to 31 December 2023). This model will
be referred to as Mod1 in the following text.
**Table 1. Estimates of the regression coefficients describing the attenuation by the cloud of erythemal DRE for three**
**ranges of noon SZA according to Eq. (6).**

| Regression Coefficients | | | | | |
|---|---|---|---|---|---|
| $\alpha$ | $\beta$ | $\alpha$ | $\beta$ | $\alpha$ | $\beta$ |
| $SZA_N < 45°$ | | $SZA_N \geq 45°$ and $<60°$ | | $SZA_N \geq 60°$ | |
| 0.954 | 0.844 | 0.918 | 0.750 | 0.960 | 0.697 |

The next two regression models were built using the monthly averages of erythemal DRE, $RE_{ERYT}(YR, M)$, for
month $M$ in year $YR$ (from 2014 up to 2023) averaging all available daily $RE_{ERYT}(D)$ values in $M$ month for $YR$



year. The corresponding long-term monthly means for $M$ month, $RE^*_{ERYT}(M)$, is from the averages of all data for
this calendar month. The idea of these models is to explain relative changes in the erythemal monthly RE, i.e.,
$\Delta ER(YR, M) = 100\% \, (RE_{ERYT}(YR, M) - RE^*_{ERYT}(M))/RE^*_{ERYT}(M)$ with the corresponding relative changes in the UV
explaining variables $X$, i.e., $\Delta X(YR, M) = 100\%(X(YR, M) - X^*(M))/X^*(M))$, where $X=\{G, TCO_3\}$:
$$\Delta ER_K(YR, M) = a_K(M)\, \Delta G(YR, M) + b_K(M)\Delta TCO_3(YR, M) + c_k , \qquad (7)$$
where K=OBS and ERA5 are for the regression using the explaining variables from the measurements at Belsk
and ERA5 reanalysis, respectively. Finally, the modelled $RE_{ERYT,K}(YR, M)$ value is equal to:
$$RE_{ERYT,K}(YR, M) = RE^*_{ERYT,K}(M)\left(1 + \frac{a_K(M)\,\Delta G(YR, M) + b_K(M)\Delta TCO_3(YR, M) + c_K}{100}\right), \qquad (8)$$
Models defined by Eq. (8) were used to compare fluctuations in UV data in periods with RB and SL501 A
measurements relative to the long-term monthly means in these periods, $RE^*_{ERYT}(M)$, that were approximated using
the long-term averages of the measured $RE_{ERYT}(D)$ values for the period 1976−1992 and 1993−2013, respectively.
The regression coefficients, $a_K$, $b_K$, and $c_K$, which were calculated using the standard least-squares linear fit to the
most reliable (2014−2023) data (Table 2), were applied to construct monthly time series for the entire measurement
period (1976−2023). The model for K=OBS and ERA5 in Eq. (8) is denoted further in the text as Mod2 and
Mod3, respectively.
**Table 2. Coefficients of the multilinear regressions derived for each calendar month based on the explaining variables**
**from the measurements at Belsk (Mod2) and ERA5 reanalysis (Mod3) data for the period 2014−2023.**

| Month: | Mod 2 | | | Mod 3 | | |
|---|---|---|---|---|---|---|
| | $a_{OBS}$ | $b_{OBS}$ | $c_{OBS}$ | $a_{ERA5}$ | $b_{ERA5}$ | $c_{ERA5}$ |
| January | 0.84 | −0.77 | −5.69 | 1.34 | −1.22 | −8.38 |
| February | 0.81 | −1.12 | −0.12 | 0.95 | −1.40 | −0.05 |
| March | 0.59 | −0.93 | −0.65 | 0.84 | −0.98 | −0.77 |
| April | 0.90 | −0.85 | −1.94 | 1.26 | −1.22 | −3.77 |
| May | 0.86 | −2.00 | 1.14 | 0.86 | −1.97 | 0.64 |
| June | 1.08 | −0.87 | −0.05 | 1.14 | −0.83 | 0.11 |
| July | 0.69 | −0.84 | 0.00 | 0.40 | −0.99 | −0.00 |
| August | 0.82 | −1.46 | −1.99 | 0.63 | −2.05 | −1.40 |
| September | 0.86 | −0.79 | −0.00 | 0.94 | −0.97 | −0.00 |
| October | 0.80 | −1.12 | −0.49 | 0.86 | −0.45 | −0.52 |
| November | 0.58 | −1.15 | −1.02 | 0.66 | −0.73 | −0.97 |
| December | 0.73 | −0.23 | 2.11 | 1.28 | 2.61 | 0.77 |

**2.4 Statistical methods**
Several standard statistical characteristics, which are calculated from the relative differences, $z_i$, between the
observed, $x_i$, and model value, $y_i$, values expressed in percentage of the observed value, are used to determine the
level of agreement between two time series. These are as follows: mean relative error (MRE), mean absolute error
(MAE), standard error (SE), root mean square error (RMSE), and Pearson's correlation coefficient (R):
$$z_i = 100\%\frac{x_i - y_i}{x_i} , \quad i = 1, \dots, N., \qquad (9)$$
$$MRE = \frac{1}{N}\sum_{i=1}^{N} z_i , \qquad (10)$$




$MAE = \frac{1}{N}\sum_{i=1}^{N}|z_i|$ ,         (11)
$SD = \left(\frac{1}{N}\sum_{i=1}^{N}(z_i - MRE)^2\right)^{\frac{1}{2}}$ ,         (12)
$RMSE = \left(\frac{1}{N}\sum_{i=1}^{N}z_i^2\right)^{\frac{1}{2}}$ ,         (13)
$R = \frac{\sum_{i=1}^{N}(x_i-<x>)(y_i-<y>)}{\left(\sum_{i=1}^{N}(x_i-<x>)^2\right)^{\frac{1}{2}}\left(\sum_{i=1}^{N}(y_i-<y>)^2\right)^{\frac{1}{2}}}$ , $<x> = \frac{1}{N}\sum_{i=1}^{N}x_i$ , $<y> = \frac{1}{N}\sum_{i=1}^{N}y_i$ ,         (14)
Standard least-squares linear regression is applied to find the long-term tendency in the data. According to
Weatherhead et al. (1998), the standard error of the linear trend estimate, $SE_{LS}$, by standard least-squares approach
should be multiplied by the factor $F = \sqrt{1 + R_{k+1})/(1 - R_{k+1}}$ to obtain the standard error corrected for the
autocorrelation (with a time lag of 1) in the trend residuals, $SE_{LS, COR}$, if the trend residuals are positively correlated
with the autocorrelation coefficient equal to $R_{k+1}$.(for $R_{k+1} < 0$, $F=1$):
$SE_{LS, COR} = F \times SE_{LS}$,         (15)
Further in the text (Section 3.3), the slopes of the regression line will be calculated by Matlab function $-fitlm(x,y)$,
and the corrected standard error of the slope, $SE_{LS, COR}$ for cases with $R_{k+1} > 0$, will be enlarged by the factor
proposed by Weatheread et al. (1998) (see Eq. (15)).
**3 Results**
**3.1 The reevaluation of the UV measurements since 5 August 2013**
On 5 August 2013, the KZ616 replaced the previous SL501 A #2011, which had been routinely used for UV
monitoring since 1995, as its performance had deteriorated (Fig. 2). Following this change, a new calibration
procedure for the Belsk's biometer data was introduced for early detection of instrument failure. Each month its
output (erythemal irradiance) was compared with the corresponding output of the collocated BS64. An example
of such a monthly comparison (for June 2023) and time series of the monthly means of the ratio between BS064
and KZ616 erythemal DRE are shown in Fig. 3a and Fig. 3b, respectively.

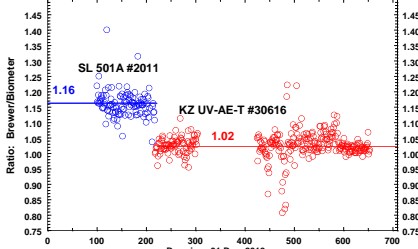

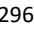

**Figure 2. The ratio between the erythemal DRE from the biometers (SL501 A #2011 before 5 August 2013 and KZ616**
**afterwards) and the Brewer Mark II spectrophotometer for the 2013−2014 period. The horizontal lines denote the mean**
**value of the ratio.**

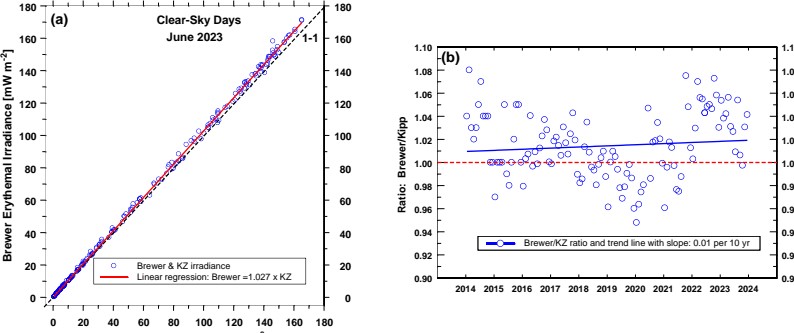

**Figure 3. Comparison of the BS64 and KZ616 erythemal data: (a) the ratio between the erythemal irradiances measured in June 2023 for clear-sky days, (b) time series of the monthly BS64/KZ616 ratios for the 2014−2023 period.**

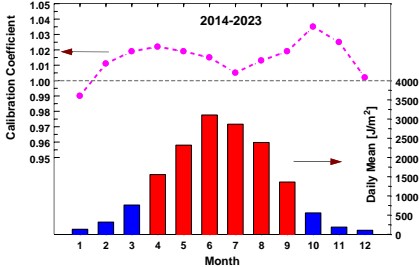

**Figure 4. Seasonal pattern of the calibration coefficient (CC ver. 1) and daily erythemal RE for the period 2014−2023. Red bars denote months contributing mostly to the annual RE.**

The long-term (2014−2023) means of the monthly CC1 and erythemal DRE for each calendar month are shown in the upper and lower graphs of Fig. 4. The *CF* values are in the range of 1.00 to 1.02 during the period (April–September) when the intensity of solar UV radiation is usually high and the fine weather often allows prolonged outdoor activity. Given this and the insignificant trend in the time series of the monthly BS64/KZ ratios (Fig. 3b), it was decided to keep the original KZ616 data without additional adjustments. This assumption is also supported by the BS64/KZ616 comparisons for all BE data considered for the period 2014−2023, as shown by the linear regressions close to the 1-1 perfect agreement line in the three scatter plots (Fig. 5). For the daily vitamin D3 and antipsoriatic RE, the values were reconstructed from the daily erythemal RE using the transfer coefficients defined by Eq. (4) (the values are archived in the IG PAS Data Portal, Krzyścin (2024), but the corresponding Brewer values were calculated from the real measured spectra weighted with the action spectra shown in Fig.1).



316

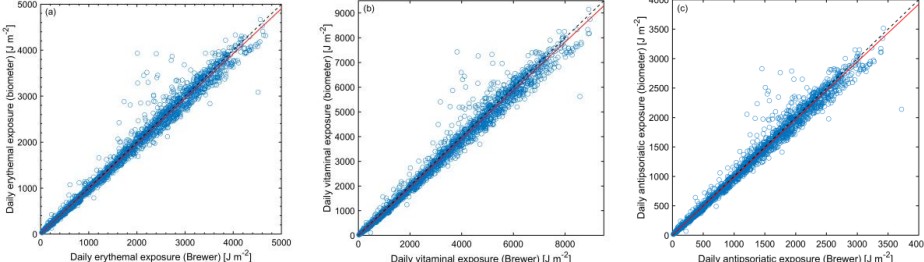

317

**Figure 5**. Scatter plots (KZ616 versus BS64) for biologically effective DRE in the period 2014−2023: (a) erythema appearance, (b) previtamin D3 synthesis, and (c) psoriasis clearance.

Table A1 shows the values of the descriptive statistics for the period 2014−2023 according to the different ranges of SZA values at noon ($SZA_N$), which confirm the good agreement between the DRE for all considered biological effects from the well-calibrated BS64 and KZ616 used in routine UV monitoring. For example, regardless of the biological effect, MRE and RMSE are ~1 % and ~9 % for $SZA_N < 45°$, which occurs from 8 April to 5 September at Belsk, i.e. during the period with the highest UV intensity of the year. For $SZA_N \geq 60°$ (from 15 October up to 27 February of next year), MRE and RMSE are only slightly larger (~2 % and ~10 %, respectively) for the erythema and antipsoriatic exposures. These values are higher (~13 % and ~18 %) for the previtamin $D_3$ exposures, raising questions about the usefulness of the erythema biometers for measuring vitamin $D_3$ exposure. However, vitamin $D_3$ synthesis in the skin ceases during this period.

**3.2 The reevaluation of the UV measurements before 5 August 2013**

**3.2.1 Calibration coefficients from the clear-sky model simulations**

Analyses of intraday UV measurements in Belsk from 1 January 1976 to 4 August 2013 have to be divided into two parts, i.e. 1 January 1976−31 December 1992, and 1 January 1993−4 August 2013, due to the different broadband instruments used for UV monitoring. In the first period, daily erythemal exposures were archived on the basis of manual summation of RB counts per day. For the latter period, 1-min erythemal irradiances were automatically recorded by a logger using SL501 A biometers and utilized in the calculation of UVI and daily erythemal RE. Two methods of data calibration for the period 1976−2013 are proposed (Sect. 2.3.2) using clear-sky data: modelled and measured daily erythemal RE and UVI for the correction method denoted CC1 and CC2, respectively. Figure 6a shows the time series of CC1 and CC2 values together with their smoothed values by the LOcally Weighted Scatterplot Smoothing (LOWESS) smoother, Cleveland (1979), which were used as multipliers of the raw UV data before 5 August 2013. The differences between CC1 and CC2 are shown in Fig. 6b.

In the former period, UVI values were not archived. This means that CC2 values cannot be directly calculated. However, CC2 values equal to 1 could be assumed as the output of the RB instrument was previously adjusted to that by SL501 A #927 using their simultaneous measurements for the period 1992−1994 (Puchalski et al., 1995). Such an assumption can be supported by a small jump (~1 %) in the differences between CC1 and CC2 values in January 1993 (Fig.6b). This jump is really small taking into account that the 1993 adjustment of RB meter was inferred from field comparisons between RB and SL501 A #927 but here this is calculated from smoothing ratios

between modelled and observed UVI for clear-sky days. Moreover, in the period 1976−1993, an oscillation with
0.015 amplitude is seen around the constant level of CC1=1.045 which justifies the assumption of an almost
constant CC2 pattern before 1993. Using two sets of the reevaluated 1976−2013 data will allow us to discuss the
robustness of trend calculations for the entire 1976−2023 period of the UV measurements at Belsk (Sect. 3.3).

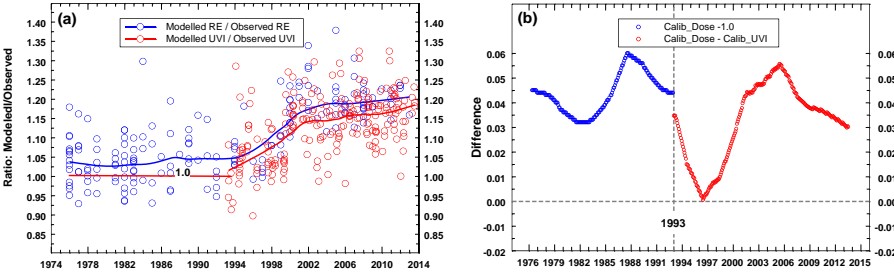


**Figure 6**. (**a**) **TUV model-observation ratios for erythemal DRE and UVI obtained for clear-sky days. The solid curves**
**represent smoothed values of the ratios to be used as the calibration coefficients, i.e., the multipliers applied to the raw**
**measurements. The multipliers were set equal to 1 for the 1976−1993 calibration based on UVI, (b) differences between**
**the monthly means of the calibration coefficients shown in Fig.6a.**

### 3.2.2 Statistical models

Erythemal DRE for the period 1 January 1976 – 4 August 2013 were reconstructed with Mod1 defined by
Eq.(5)−(6). The model's constants came from the model training using the original KZ data and the explaining
variables (TCO$_3$ and CI) from 5 August 2013 − 31 December 2023 period. The reconstructed values were
compared with two sets of the reevaluated data obtained before 5 August 2013 after multiplying raw daily
erythemal RE with CC1 and CC2, respectively. Figure 7 shows the scatter plot of the reconstructed versus
reevaluated erythemal DRE with CC1 (Fig.7a) and CC2 (Fig.7b) multipliers of the raw data for the three periods
corresponding to the RB, SL501A, and KZ616 measurements, respectively. The points in Figure 7 cluster around
a line of perfect 1-1 agreement with only a few outliers. It seems that there is only a small difference between the
reevaluated daily erythemal RE and the corresponding output of Mod1 with the CC1 and CC2 multipliers. This is
also supported by similar values of the descriptive statistics for the periods 1976−1992 and 1993−2013 (Table 3).
It is worth mentioning that the performance of Mod1 resembles that of the Brewer spectrophotometer from the
comparison with the original KZ616 data (see almost the same values of descriptive statistics for the full-year data
in Table 3 and Table A1 for the 'All SZA$_N$' cell, for example, RMSE equal to 10.5 % and 8.9 %, respectively).

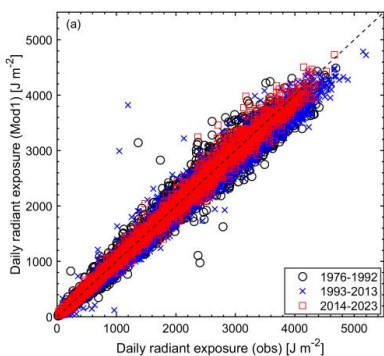
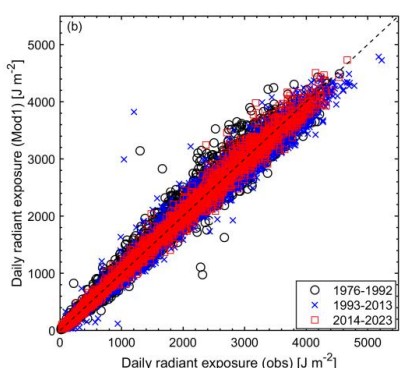

**Figure 7. Scatter plot of the modelled (Mod1) erythemal DRE versus the reevaluated observed values for the 1976–1992, 1993–2013, and 2014–2023 period, respectively: (a) CC1 version of the calibration coefficients for the period 1 January 1976–4 August 2013, (b) corresponding CC2 version of the calibration coefficients. KZ616 measurements were taken without corrections.**

Table 3. The descriptive statistics (MRE, MAE, RMSE, and SD, as defined in Sect. 2.4) calculated from the relative daily differences, 100% (reevaluated measurement − Mod1 value)/(reevaluated measurement), for the periods 1976–1992, 1993–2013 and 2014–2023. The correlation coefficient R was obtained from the reevaluated measurements and modelled values. Two versions of the reevaluated datasets were considered, using CC1 and CC2 multipliers on the raw measurements. Both datasets include raw KZ616 data as there was no need to recalculate these data. The results are shown for annual (January–December) and summer (June–August) data.

| Statistics | Year-Round (January−...−December) | | | | | June−July−August | | | | |
|---|---|---|---|---|---|---|---|---|---|---|
| | Multipliers of the raw data | | | | | | | | | |
| | 1976−1992 | | 1993−2013 | | 2014−2023 | 1976−1992 | | 1993−2013 | | 2014−2023 |
| | CC1 | CC2 | CC1 | CC2 | CC=1 | CC1 | CC2 | CC1 | CC2 | CC=1 |
| MRE | 2.7 | −1.6 | 1.9 | 0.3 | −1.4 | 0.8 | −3.5 | 2.6 | 1.0 | −0.9 |
| MAE | 9.8 | 9.6 | 9.7 | 9.4 | 6.8 | 7.8 | 8.1 | 7.0 | 6.4 | 5.2 |
| RMSE | 13.7 | 14.1 | 14.5 | 14.6 | 10.5 | 10.8 | 11.7 | 10.1 | 9.7 | 6.9 |
| R | 1.00 | 0.99 | 1.00 | 0.99 | 1.00 | 0.96 | 0.96 | 0.97 | 0.98 | 0.98 |
| SD | 13.4 | 14.0 | 14.3 | 14.6 | 10.4 | 10.9 | 11.2 | 9.8 | 9.7 | 6.8 |

Erythemal DRE by Mod1 can be obtained for days when the explanatory variables, $TCO_3$ and CI, are available from the collocated measurements at Belsk by the Dobson radiometer and pyranometer, respectively. It is therefore possible to fill gaps in the measured data and obtain a complete (1976−2023) series of erythemal DRE to be used in calculations of erythemal annual and summer (June−July−August) RE. These data can also be calculated using the erythemal monthly RE based on Mod2 and Mod3. All these series are analysed in section 3.3 for trend calculations to assess the level of uncertainty in the long-term variability of the Belsk UV data.

Table 4 shows the values of the descriptive statistics for the three models used (Mod1, Mod2 and Mod3) and two versions of the reevaluated data (using CC1 and CC2 multipliers on the raw data) based on the annual and summer RE. The differences between descriptive statistics (MRE, MAE, RMSE, SD) in CC1 and CC2 columns are within a few percentage points for MRE and about 1−1.5 percentage points for other statistics, indicating that the two independent calibration methods give fairly similar results. The performance of Mod2 and Mod3 is in most cases slightly better than that of Mod1 (Table 4) because these models add fluctuations to the mean values for the periods 1976−1992, 1993−2013 and 2014−2023 calculated from the reevaluated measurements of RB, SL501A (#919 and #2011 for the periods 1993−1994 and 1995−2013 respectively) and the original KZ616 measurements.

All models considered were designed to test whether changes in the primary UV drivers, ozone and clouds, explain
year-to-year UV variability. The performance of Mod3 is surprisingly similar to that obtained from Mod2 despite
the use of UV proxies from the ERA5 reanalysis. This confirms the possibility of using explanatory variables from
these reanalyses to fill gaps in the proxy data.
The lowest correlation coefficients between the reevaluated measurements and modelled values were found in the
period 1993−2013 for the measurement-model pairs with the same version of the CC multipliers (CC1 or CC2).
This is particularly pronounced for the summer data (see e.g. Mod3 values of 0.50 and 0.43 for CC1 and CC2
pairs, respectively), suggesting a poorer agreement between measurements and model in the period 1993−2013.
This was found for all models. However, other descriptive statistics (MRE, MAE, RMSE and SD) differed only
slightly, i.e. less than 1.5 percentage points.
**Table 4. Same as Table 3, but the descriptive statistics are calculated using time series of erythemal annual and summer**
**RE.**

| | Year-Round: January−…−December | | | | | Summer: June−July−August | | | | |
| Statistics | Multipliers of the raw data | | | | | | | | | |
| | 1976−1992 | | 1993−2013 | | 2014−2023 | 1976−1992 | | 1993−2013 | | 2014−2023 |
| | CC1 | CC2 | CC1 | CC2 | CC=1 | CC1 | CC2 | CC1 | CC2 | CC=1 |
| | | | | | Mod1 | | | | | |
| MRE | 3.3 | −0.9 | 4.2 | 2.6 | 0.2 | 1.9 | −2.4 | 3.5 | 1.9 | −0.5 |
| MAE | 3.9 | 2.5 | 4.7 | 3.2 | 1.0 | 4.0 | 3.4 | 4.4 | 2.8 | 1.8 |
| RMSE | 4.4 | 2.9 | 5.0 | 3.5 | 1.2 | 4.4 | 4.5 | 4.8 | 3.4 | 2.6 |
| R | 0.82 | 0.86 | 0.77 | 0.83 | 0.93 | 0.92 | 0.93 | 0.57 | 0.65 | 0.96 |
| SD | 3.2 | 3.0 | 2.8 | 2.4 | 1.4 | 4.2 | 4.0 | 3.3 | 2.9 | 2.7 |
| | | | | | Mod2 | | | | | |
| MRE | 0.9 | 1.0 | 0.5 | 0.6 | 0.3 | 1.0 | 1.1 | 0.6 | 0.6 | −0.0 |
| MAE | 2.1 | 2.0 | 2.0 | 1.8 | 0.6 | 3.3 | 2.9 | 2.2 | 1.9 | 1.3 |
| RMSE | 2.6 | 2.4 | 2.7 | 2.3 | 0.8 | 4.0 | 3.7 | 2.8 | 2.4 | 1.8 |
| R | 0.90 | 0.92 | 0.81 | 0.86 | 0.97 | 0.93 | 0.94 | 0.72 | 0.79 | 0.98 |
| SD | 2.6 | 2.4 | 2.7 | 2.3 | 0.8 | 4.1 | 3.7 | 2.8 | 2.4 | 1.9 |
| | | | | | Mod3 | | | | | |
| MRE | 0.4 | 0.5 | 0.9 | 0.9 | 0.6 | −0.2 | −0.1 | 0.3 | 0.3 | −0.1 |
| MAE | 1.5 | 1.4 | 2.7 | 2.8 | 0.8 | 3.0 | 3.2 | 2.9 | 2.9 | 2.1 |
| RMSE | 1.7 | 1.9 | 3.4 | 3.6 | 0.9 | 3.7 | 3.7 | 3.7 | 3.7 | 2.5 |
| R | 0.96 | 0.94 | 0.70 | 0.67 | 0.97 | 0.94 | 0.94 | 0.50 | 0.43 | 0.92 |
| SD | 1.8 | 2.0 | 3.4 | 3.5 | 0.8 | 3.9 | 3.9 | 3.8 | 3.8 | 2.7 |

**3.3 Trend analyses**
**3.3.1 The erythemal annual and summer radiant exposures in the period 1976-2023**
Trend analyses are applied to the erythemal annual and summer RE based on daily (for reevaluated observations
with filled gaps, $OBS_F$, and Mod1) and monthly RE (for Mod2 and Mod3). Gaps in the measurements were filled
using Mod1 simulations. Two versions of the $OBS_F$, Mod2 and Mod3 time series are possible because of the use
of CC1 and CC2 multipliers on the raw (1976−2013) daily measurements. For the Mod1 time series, only one
series was available for analysis, as this model reconstructed erythemal RE using the proxy values and the model
coefficients estimated from the KZ616 measurements (2014−2023), which did not require calibration.
For Mod2 and Mod3, two variants of the time series were available as these models required the 1976−1992 and
1993−2013 mean values taken from the reevaluated measurements with two possible options (CC1 or CC2) for





the calibration multipliers. The 1976−2023 time series for the erythemal annual and summer RE using CC1 and
CC2 calibration multipliers are shown in Fig. 8 and Fig. A1, respectively. Fig.8a (Fig.A1a) and Fig.8b (Fig.A1b)
are for the erythemal annual (and summer) RE.
Linear regression lines are superimposed on the graphs to illustrate the long-term variability in the time series.
Two independent lines are drawn to account for a change in the trend pattern observed in the time series somewhere
in the early 2000s. The year of the trend change was calculated by examining the performance of fifteen
combinations of this two-line pattern, varying the year of the trend change point (from 1995 to 2009). The best fit
with maximum determination coefficients was found for the trend change point in 2005. Therefore, the slopes of
the regression lines (in kJ m$^{-2}$ per year) and the trend values (in % per year) shown in Table 5 and Table 6,
respectively, are calculated for the 1976−2004 and 2005−2023 periods. Standard deviations of the trend estimates
are calculated according to Eq. (15) if the consecutive values in the trend residuals are positively correlated, i.e.
the autocorrelation coefficient $R_{k+1} > 0$ (also shown in Tables 4−5).
The interannual variations and trend lines of erythemal annual RE are close to each other when comparing the
upper graphs in Fig. 8 and Fig.A1. This can also be observed for the summers when comparing the corresponding
lower plots. At the beginning of the RB observations (1976−1986), there were large oscillations from year to year,
suggesting an instrumental problem with the data. However, all modelled time series show quite similar
oscillations for this period, supporting the thesis that a specific combination of $TCO_3$ and cloud transparency may
be responsible for such oscillations.

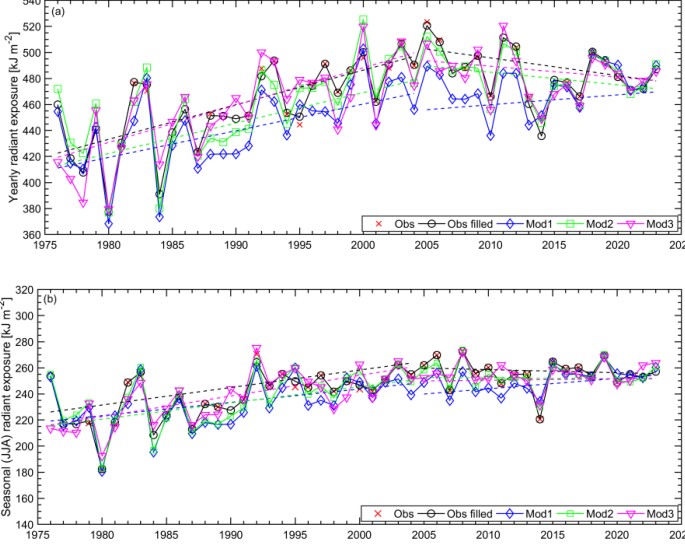


**Figure 8. Time series (1976−2023) of the erythemal radiant exposures from reevaluated observations (Obs), reevaluated observations with filled gaps (Obs filled), and model estimates (Mod1, Mod2, and Mod3) using the CC1 version of the calibration coefficients: (a) annual (January−December) exposures; (b) summer (June−Aug) exposures. Dashed lines represent the linear trends calculated for the period 1976−2004 and 2005−2023.**





The slopes of the linear fit to the analysed time series (Table 5) show a statistically significant positive trend
between 1976 and 2004 of around 20−30 kJ m$^{-2}$ and 10−20 kJ m$^{-2}$ per decade in the annual and summer data,
respectively. The trends are mostly insignificant for the period 2005−2023, with only one exception (for the Mod1
data) with a continued positive trend of ~ 10 kJ m$^{-2}$ per decade. The corresponding trend values expressed in
dimensionless units (Table 6) have the same values of about 4−7% per 10 years in the former period for both the
annual and summer time series. In the latter period, the positive trend of Mod1 is ~3 % per 10 years. The smallest
and the largest trends are always provided by Mod1 and Mod3 with CC2 calibration coefficients. However, the
differences between these trends are within the range of ± 2 standard errors of the trend estimates, taking into
account the autocorrelation in the residuals of the models (column $SE_{LS, COR}$ in Table 5).
By averaging all available statistically significant annual and summer trend values shown in the third and seventh
columns of Table 5 and Table 6, the following trends and their standard errors are obtained: for the period
1976−2004: 27.4 ± 4.4 kJ m$^{-2}$ and 5.64 ± 0.92 % per decade for the erythemal annual RE, and 14.3 ± 4.3 kJ m$^{-2}$
and 5.63 ± 1.03 % per decade for the erythemal summer RE. These values correspond to the average trend from
the two series based only on the reevaluated measurements (OBS$_F$ values in the Tables), i.e. 28.7 kJ m$^{-2}$
and 5.9 % per decade for the erythemal annual RE, and 14.3 kJ m$^{-2}$ and 5.6 % per decade for the erythemal
summer RE.
**Table 5. Trends (kJ m$^{-2}$ per year) by the linear least-squares fit to the time series of erythemal annual and summer**
**radiant exposures shown in Fig.8 and Fig.A1 calculated for the periods 1976−2004 and 2005−2023.** *SE$_{LS, COR}$* **denotes the**
**standard error of the trend estimate taking into account the autocorrelation (with a lag of 1 year) in the series of the**
**residuals of the trend model.** *R$_{k+1}$* **denotes the correlation coefficient in the lagged residuals. Bold font indicates a**
**statistically significant trend value at the 2-sigma level.**

| Data Type | Correct. Method | Annual (January...−...December) sum [kJ m$^{-2}$] | | | | Summer (June−July−August) sum [kJ m$^{-2}$] | | | |
|---|---|---|---|---|---|---|---|---|---|
| | | Trends$_{1976-2004}$ | | Trends$_{2005-2023}$ | | Trends$_{1976-2004}$ | | Trends$_{2005-2023}$ | |
| | | Trend ± SE$_{LS,COR}$ | R$_{k+1}$ | Trend ± SE$_{LS,COR}$ | R$_{k+1}$ | Trend ± SE$_{LS,COR}$ | R$_{k+1}$ | Trend ± SE$_{LS,COR}$ | R$_{k+1}$ |
| OBS$_F$ | CC1 | **2.66 ±0.52** | −0.11 | −1.36 ±0.98 | 0.17 | **1.34 ±0.37** | 0.08 | −0.24 ±0.49 | −0.32 |
| | CC2 | **3.08 ±0.52** | −0.06 | −0.45 ±0.87 | 0.14 | **1.52 ±0.37** | 0.07 | −0.26 ±0.48 | −0.30 |
| Mod1 | − | **2.05 ±0.57** | −0.19 | 0.76 ±0.78 | 0.08 | **1.02 ±0.38** | −0.13 | **0.80 ±0.36** | −0.20 |
| Mod2 | CC1 | **2.34 ±0.61** | −0.16 | −0.97 ±0.85 | 0.12 | **1.24 ±0.39** | −0.08 | −0.06 ±0.41 | −0.38 |
| | CC2 | **2.84 ±0.61** | −0.10 | −0.30 ±0.79 | 0.10 | **1.50 ±0.39** | −0.01 | 0.29 ±0.41 | −0.32 |
| Mod3 | CC1 | **2.84 ±0.56** | −0.21 | −0.84 ±0.76 | −0.08 | **1.58 ±0.32** | 0.02 | 0.11 ±0.37 | −0.13 |
| | CC2 | **3.34 ±0.54** | −0.22 | −0.17 ±0.72 | −0.13 | **1.82 ±0.20** | 0.05 | 0.46 ±0.36 | −0.13 |


**Table 6. Same as Table 5, but the results are for the trend values expressed in % per year.**

| Data Type | Correct. Method | Annual (January..−..December) sum [% yr$^{-1}$] | | | | Summer (June−July−August) sum [% yr$^{-1}$] | | | |
|---|---|---|---|---|---|---|---|---|---|
| | | Trends$_{1976-2004}$ | | Trends$_{2005-2023}$ | | Trends$_{1976-2004}$ | | Trends$_{2005-2023}$ | |
| | | Trend ± SE$_{LS,COR}$ | R$_{k+1}$ | Trend ± SE$_{LS,COR}$ | R$_{k+1}$ | Trend ± SE$_{LS,COR}$ | R$_{k+1}$ | Trend ± SE$_{LS,COR}$ | R$_{k+1}$ |
| OBS$_F$ | CC1 | **0.54 ±0.11** | −0.11 | −0.28 ±0.17 | 0.17 | **0.52 ±0.14** | 0.08 | −0.09 ±0.19 | −0.32 |
| | CC2 | **0.64 ±0.11** | −0.06 | −0.09 ±0.16 | 0.14 | **0.60 ±0.14** | 0.07 | 0.10 ±0.19 | −0.30 |
| Mod1 | − | **0.42 ±0.12** | −0.19 | 0.16 ±0.15 | 0.08 | **0.40 ±0.15** | −0.13 | **0.31 ±0.14** | −0.20 |
| Mod2 | CC1 | **0.48 ±0.13** | −0.16 | −0.20 ±0.15 | 0.12 | **0.49 ±0.15** | −0.08 | −0.02 ±0.16 | −0.38 |
| | CC2 | **0.59 ±0.12** | −0.10 | −0.06 ±0.15 | 0.10 | **0.59 ±0.15** | −0.01 | 0.11 ±0.16 | −0.32 |
| Mod3 | CC1 | **0.59 ±0.12** | −0.21 | −0.17 ±0.16 | −0.08 | **0.62 ±0.12** | 0.02 | 0.04 ±0.15 | −0.13 |
| | CC2 | **0.69 ±0.11** | −0.22 | −0.04 ±0.15 | −0.13 | **0.72 ±0.13** | 0.05 | 0.18 ±0.14 | −0.13 |

**3.3.2 The vitamin D3 and antipsoriatic annual and summer radiant exposures in the period 1976-2023**
The standard biometer used to monitor erythemal irradiance can also measure non-erythemal irradiance
(Czerwińska and Krzyścin, 2024a). Figure 5 and Table A1 provide that the daily vitamin D3 and antipsoriatic RE
derived from the KZ616 measurements agree with the directly measured BS64 values in the same way as the
original (erythemal) KZ616 data. This supports the method of the transfer from erythemal irradiance to non-
erythemal irradiance proposed by Czerwińska and Krzyścin (2024a).
Figure 9 shows the time series of the annual and summer values of the previtamin D3 synthesis and psoriasis
healing RE from 1976 to 2023. It looks like these time series are very similar when comparing the vitamin D3 to
the antipsoriatic time series. Moreover, these time series are similar to the erythemal series shown in Fig.8. The
correlation coefficients between the pairs of time series shown in Fig.8 and Fig.9, i.e. erythema & vitamin D3,
erythema & psoriasis, vitamin D3 & psoriasis, are in the range [0.90, >0.999] with the smallest value for the cases
of erythema & vitamin D3, erythema & psoriasis calculations using the summer data from Mod1 simulations.

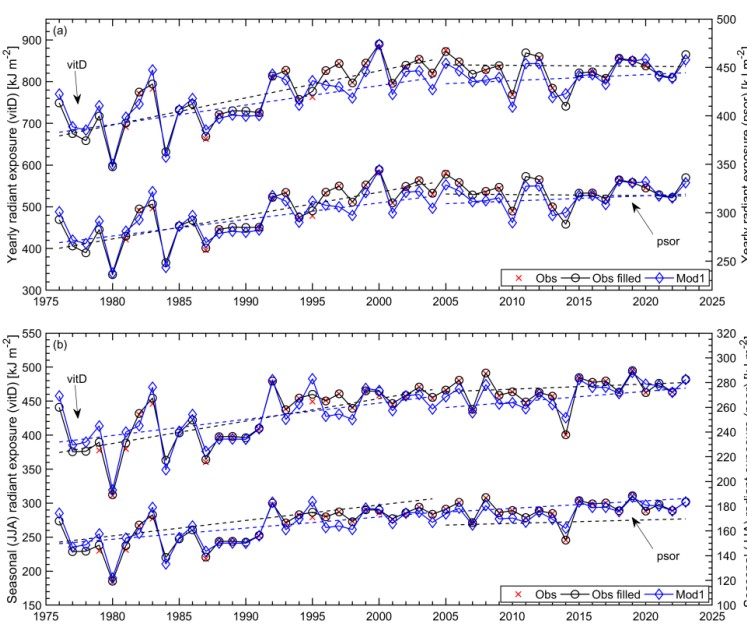

**Figure 9. Time series (1976−2023) of the previtamin D3 synthesis and psoriasis healing radiant exposures from**
**reevaluated observations (Obs), reevaluated observations with filled gaps (Obs filled), and model Mod1 estimates**
**(Mod1) using the CC2 version of the calibration coefficients: (a) annual (January−December) exposures; (b) summer**
**(June-July-August) exposures. Dashed lines represent the linear trends calculated for the period 1976−2004 and**
**2005−2023.**
Table 7 shows the trend values for the period 1976-2004 and 2005-2023 from the time series calculated using the
erythemal DRE multiplied by the transfer coefficients defined by Eq. (4). The transfer coefficients depend on only
two parameters ($TCO_3$ and SZA), even on cloudy days, as previously shown by Czerwińska and Krzyścin (2024a).
The statistically significant trend values for previtamin D3 synthesis and psoriasis clearance are slightly higher,
by about 1−1.5 percentage points per decade, than the corresponding trend values for the erythema appearance



shown in Table 6. Taking into account the standard error of the trend estimate of about 1% per decade, it cannot
be said that the differences between the trends are statistically significant.
**Table 7. Same as Table 6, but trend values are for previtamin D3 synthesis and psoriasis clearance.**

| Data Type | Correct. Method | Annual (January−December) RE [% per year] | | | | Summer (June−July−August) RE [% per year] | | | |
|---|---|---|---|---|---|---|---|---|---|
| | | Trends$_{1976-2004}$ | | Trends$_{2005-2023}$ | | Trends$_{1976-2004}$ | | Trends$_{2005-2023}$ | |
| | | Trend ± SE$_{LS, COR}$ | R$_{k+1}$ | Trend ± SE$_{LS, COR}$ | R$_{k+1}$ | Trend ± SE$_{LS, COR}$ | R$_{k+1}$ | Trend ± SE$_{LS, COR}$ | R$_{k+1}$ |
| | | Previtamin D3 synthesis | | | | | | | |
| OBS$_F$ | CC1 | **0.70 ±0.12** | −0.12 | −0.27 ±0.22 | 0.12 | **0.64 ±0.16** | 0.06 | −0.07 ±0.20 | −0.25 |
| | CC2 | **0.77 ±0.12** | −0.07 | −0.03 ±0.19 | 0.08 | **0.71 ±0.15** | 0.05 | 0.16 ±0.19 | −0.32 |
| Mod1 | − | **0.56 ±0.14** | −0.20 | 0.17 ±0.16 | 0.02 | **0.51 ±0.16** | −0.15 | **0.34 ±0.14** | −0.18 |
| | | Psoriasis clearance | | | | | | | |
| OBS$_F$ | CC1 | **0.66 ±0.12** | −0.13 | −0.27 ±0.21 | 0.15 | **0.63 ±0.15** | 0.06 | −0.07 ±0.20 | −0.25 |
| | CC2 | **0.74 ±0.12** | −0.08 | −0.03 ±0.20 | 0.10 | **0.70 ±0.15** | 0.05 | 0.16 ±0.19 | −0.26 |
| Mod1 | − | **0.53 ±0.13** | −0.20 | 0.17 ±0.18 | 0.14 | **0.51 ±0.16** | −0.15 | **0.34 ±0.14** | -0.18 |


**4 Summary and Discussion**
One of the world's longest measurements of solar UV radiation at the Earth's surface (and probably the longest
taken by erythemal biometers) comes from Belsk. Measurements began in 1975 and continuous monitoring started
on 1 January 1976. To the authors' knowledge, the longest UV monitoring series began in Moscow in 1968 with a
broadband (300−380 nm) instrument developed at the Moscow State University Meteorological Observatory
(Chubarova et al., 2000).
Several biometers participated in UV monitoring at Belsk, starting with RB, which operated until 31 December
1992. Subsequently, biometers SL501 A (#919 and #2011) and, since 5 August 2013, KZ616 have participated in
UV monitoring. Each of these instruments has individual characteristics (spectral response, cosine error, ageing
rate) and technical solutions, e.g. RB was not temperature stabilised and its output was in solar burn units.
Therefore, a retrospective re-evaluation of the Belsk UV time series was necessary, and the homogenisation of the
data from 1976 to 2023 is presented in this article.
Belsk is a unique observatory where UV monitoring has been accompanied by monitoring of ozone (TCO$_3$),
aerosol optical properties (AOD) and cloud characteristics (sunshine duration, CI from global solar irradiance
measurements), i.e. basic input parameters to a radiative transfer model allowing reconstruction of the erythemal
RE. In addition, collocated BS64 measurements of UV spectra allow monthly verification of actual KZ616
performance. BS64 spectral measurements also allow assessment of the quality of Czerwińska and Krzyścin
(2024a) retrieval to convert standard erythemal measurements to the non-erythemal BE irradiance (see the cases
of the vitamin D3 and antipsoriatic DRE in Figure 5).
Model simulations of erythemal DRE and UVI under cloudless sky provide a basis for the correction procedure of
raw UV data. A selection of clear-sky conditions throughout the entire day from the daily proxy values (relative
sunshine duration and RE from global solar irradiation), which were available for Belsk, is not straightforward as
only the examination of the daily course of these measurements would allow to capture cloudless moments within
the day. Therefore, two very different calibration configurations (CC1 and CC2 as defined in section 2.3.2) have
been proposed to assess the uncertainty range of the calibration method. The reevaluated time series appear quite

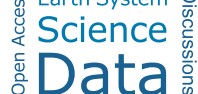

similar, i.e. the difference between these series is within a few percentage points (Fig.6). There was no need for
reevaluation of the KS616 data for the period 2014−2023 as shown by the comparisons with BS64 data (Fig.3 and
Fig. 5).
Statistical models trained on the KZ616 data for the period 2014−2023 allowed the data to be reconstructed from
the beginning of UV observations at Belsk. These reconstructed series allowed independent examination of the
pattern of interannual variability (which was unexpectedly large before 1985) and trends in erythemal annual and
summer RE. The statistical models generally mimic the observed long-term variability in the reevaluated daily
erythemal exposures. The statistically significant trend of ~6 % per decade with a standard error of ~1 % per
decade for the period 1976−2005 can be calculated (for both erythemal annual and summer RE) by averaging
trends from the sample of seven versions of trend estimates from reevaluated and reconstructed data. All individual
trend values are within the range of the mean trend ±2 standard error (i.e. there is no outlier in this trend sample).
The standard errors for the individual trend estimates are in the range of 1-1.5% per decade, i.e. quite close to the
standard error of the averaged trend derived from the trend sample. This supports the robustness of the trend
estimates in annual and summer RE for the 1976−2005 parts of the Belsk time series. In addition, it also appears
that the very different calibration methods applied to the 1976−2013 raw UV data, based on the comparisons of
clear-sky erythemal DRE (CC1 method) and UVI (CC2 method), lead to differences in the individual 1976−2005
trend estimates of about 1 % per decade (see Table 6 for the trend differences between pairs of $OBS_F$, Mod2 and
Mod3 calculated with the CC1 and CC2 correction applied to the raw time series).
We found that Mod1 could provide reasonable estimates of DRE for all biological effects considered (erythema,
vitamin D3 and psoriasis), i.e. with a bias of less than 2 % and a standard deviation of ~ 9 % (Table A1) for the
part of the year when UV radiation is of particular interest, when the midday SZA is less than 45° (i.e. below the
shadow length), according to the so-called shadow rule for protection against high UV (Downham, 1998).
Krzyścin et al. (2011) found a trend of 5.6 % ± 0.9 % (1σ) per decade in the erythemal annual RE for the period
1976−2008. This is in good agreement with the present trend estimate, regardless of the different calibration
methods used. The correction of the SL501 A data carried out in 2011 was based on simultaneous measurements
with KZ616 for the period 2008−2009 and further corrections for the instrument ageing using TUV cloudless sky
simulations.
Similar trend estimates for erythemal radiation can be inferred from the reconstructed erythemal time series for
the Moscow region based on the UV measurements by the broadband (300−380 nm) radiometer (Chubarova et al.,
2018) and the statistically reconstructed erythemal radiation series for Hradec Kralowe (Čížková et al., 2018). For
the Moscow region, the authors reported a statistically significant positive trend of more than 5 % per decade for
the period 1979-2015. Volpert and Chubarowa (2021) revealed the decadal trend in the reconstructed erythemal
UV irradiance over the Moscow region for the warm season (May−September) of 5.1 % ± 1.1 % per decade in the
period 1979−2016. Estimates from the smoothed pattern of annual erythemal exposures taken from Fig. 2c by
Čížková et al. (2018) for 1976 (~1.20 kJ m$^{-2}$ for the annual mean of erythemal daily RE) and 2005 (~1.40 kJ m$^{-2}$)
give a trend of ~5% per decade for the period 1976−2004. From around 2005, both time series show a levelling
off. Trends calculated here from the RE time series for other biological effects (previtamin D3 synthesis and
psoriasis lesion clearance), using an approach analogous to that used for the erythema data, show very similar
trends.



**5 Code and data availability**

All data have been published as free access TXT files and are made available through PANGEA repository at https://doi.org/10.1594/PANGAEA.972139 (Krzyścin et al., 2024) and IG PAS Data Portal repository: https://doi.org/10.25171/InstGeoph_PAS_IGData_Biologically_Effective_Solar_Radiation_Belsk_1976_2023 (Krzyścin, 2024). ERA5 data are publicly accessible at https://cds.climate.copernicus.eu/datasets/reanalysis-era5-single-levels?tab=overview (ERA5, 2024). MERRA-2 data are accessible at https://doi.org/10.5067/Q9QMY5PBNV1T (GMAO, 2024). Coefficients of the linear regression are calculated by Matlab function (Matlab R2018a) – $fitlm(x,y)$.

**6 Conclusions**

It is widely accepted that the use of overlapping measurement series from different instruments increases the reliability of results obtained from single time series analyses. Consequently, the inclusion of at least two different time series for analyses of the variability of a selected quantity over the entire measurement period is also beneficial for assessing data quality and establishing confidence in the results obtained. This is illustrated by the current data archived in the PANGEA (Krzyścin et al., 2024) and IG PAN Data Portal (Krzyścin, 2024). The daily characteristics of BE radiation at Belsk allow the elaboration of scenarios of human outdoor activities to obtain maximum health benefits from sunbathing while minimising the risk of erythemal overexposure. The long-term variability of erythemal radiation calculated for Belsk corresponds to that previously recorded at distant stations in central/eastern Europe, making these scenarios applicable to wider areas.

**Appendix A**

Table A1 presents descriptive statistics (defined in Sect. 2.4) of the differences between biologically effective DRE measured by the BS64 and the KZ616, $100\%(RE_{EFF,\,BS64} - RE_{EFF,\,KZ616})/RE_{EFF,BR64}$. The vitamin D3 (VitD) and antipsoriatic (Psor) RE were reconstructed from the erythemal (Eryt) RE (Sect. 2.3.3), but the Brewer RE values were calculated using the daily integral of the measured spectral irradiance weighted by the action spectra (Fig.1).

**Table A1. Descriptive statistics of the 2014-2023 differences between the daily biologically effective radiant exposure with the Brewer spectrophotometer #064 and the Kipp & Zonen erythema biometer (UV-S-AE-T #30616) at Belsk for the different midday SZA ranges (SZAN) used in the Mod1 setup.**

| Statistics | SZA$_N$<45$^{oo}$ | | | SZA$_N$ [45$^o$, 60$^o$] | | | SZA$_N$≥60$^o$ | | | All SZA$_N$ | | |
|---|---|---|---|---|---|---|---|---|---|---|---|---|
| | Eryt | VitD | Psor | Eryt | VitD | Psor | Eryt | VitD | Psor | Eryt | VitD | Psor |
| MRE | 0.6 | 1.5 | 0.7 | 2.5 | 6.6 | 3.3 | 1.7 | 13.2 | 1.7 | 1.4 | 6.8 | 1.6 |
| MAE | 5.3 | 6.0 | 5.6 | 4.9 | 7.9 | 5.4 | 6.8 | 14.7 | 7.0 | 5.8 | 9.6 | 6.1 |
| RMSE% | 8.7 | 9.2 | 9.0 | 7.2 | 10.0 | 7.8 | 10.0 | 16.3 | 10.3 | 8.9 | 12.4 | 9.3 |
| SD% | 8.7 | 9.1 | 9.0 | 6.8 | 7.6 | 7.1 | 9.9 | 9.5 | 10.2 | 8.8 | 10.4 | 9.1 |


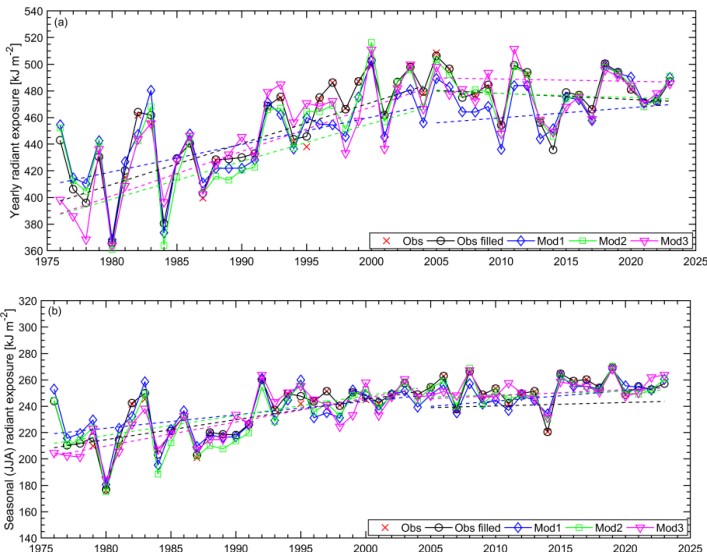

**Figure A1. Same as Fig.8, but for the reevaluated observations with the CC2 correction coefficients.**

**Authors contributions**. Conceptualisation, JK and AC; methodology, JK, AC, JJ, PS, and BR; validation, AC
and IP; visualisation, AC; writing (original draft preparation), JK and AC; writing (review and editing), JK, AC,
PS, and IP; funding acquisition, JK and JJ. All authors have read and agreed to the published version of the paper.
**Competing interests**. The contact author has declared that none of the authors has any competing interests.
**Acknowledgments**. This work was partially financially by the Chief Inspectorate of Environment Protection,
contract number GIOŚ/31/2023/DMŚ/NFOŚ.

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
