# Peer review of "Biologically effective daily radiant exposure for erythema 1"

_Earth System Science Data, 2024_

## Author Comment (AC1)

**Response to the Reviewer 1 comments and suggestions**

The following statement (here in italics) summarises the opinion of reviewer 1 in the 'General comments' section of his review of the manuscript:

"In general, the presentation of the manuscript is clear and logical. Overall, the manuscript is well written and informative, deserving for publication."

Thus, the authors' response to reviewer 1's opinion includes responses to specific/technical comments, which in his words are ""Specific and minor comments on formal aspects, mainly asking for explanations and for improving readability". These are as follow

1. Consider including, in the introduction, and if possible, a short revision for long available series of UV and erythemal data. The reference to Chubarova et al. (2000), which appears in Section 4, can be included there.

Responding to this comment we add following text to the introduction

"Long-term series of surface UV radiation from ground-based observations with a length of at least a few decades are rare. To the authors' knowledge, the longest UV monitoring series began in Moscow in 1968 with a broadband (300–380 nm) instrument developed at the Moscow State University Meteorological Observatory (Chubarova et al., 2000). One of the world's longest measurements of solar UV radiation at the Earth's surface (and probably the longest taken by erythemal broadband instruments) comes from Belsk. Measurements began in 1975 and continuous monitoring started on 1 January 1976. From a global perspective, the first UV monitoring results appeared at the World Ozone and Radiation Data Centre (WOUDC) in 1989, but continuous time series over three decades are only available for a limited number of stations including: Uccle (Belgium), Edmonton, Resolute, Toronto, Churchill, Saturna Island (Canada), Tateno (Japan) and Syowa (Antarctica) (WOUDC 2025). Database Network for the Detection of Atmospheric Composition Change (NDACC) include also stations with at least of three decades of measurements such as Lauder (New Zealand), Mauna Loa (USA) and three Antarctica stations – Arrival Heights, Palmer Station and South-Pole (NDACC 2025)." L. 86-97

2. The different periods and instruments operating along time are repeated several times along the text. Would it be possible to present a table schematising the time periods, instruments, references, the ancillary information used, and methods/models applied?

A new Table 1 has been added to the text explaining the instruments and data used in the manuscript. L.172

| Data                 | Instrument/data                   | Operation period | Reference              |
|----------------------|-----------------------------------|------------------|------------------------|
| Daily ERE and UV     | Robertson Berger Meter            | 1976-1994        |                        |
| Index                | SL Biometer 501 A # 927           | 1992-1994        | Krzyścin et al. (2024) |
|                      | SL Biometer 501 A # 2011          | 1995-2013        | Krzyścin (2024)        |
|                      | Kipp-Zonen UV-AE-T # 30616        | 2013-present     |                        |
| TCO 3     | Dobson Spectrophotometer # 84     | 1963-present     | Krzyścin (2024)        |
| SunDur               | Campbell-Stokes sunshine recorder | 1966–1968,       |                        |
|                      |                                   | 1970–1973,       | Krzyścin (2024)        |
|                      |                                   | 1975-present     |                        |
| G                    | Kipp CM 6                         | 1965-1980        |                        |
|                      | Sonntag PRM-2                     | 1981-1987        |                        |
|                      | Kipp&Zonen CM 5                   | 1988-1991        | Krzyścin (2024)        |
|                      | Kipp&Zonen CM 11                  | 1992-2010        |                        |
|                      | Kipp&Zonen CM 21                  | 2010-present     |                        |
| AOD 340nm | Sonntag pyrheliometers            | 1976-2013        | Krzyścin (2024)        |
|                      | CIMEL CE 318-T                    | 2004-present     |                        |
| G and G 0 | ERA5 reanalysis                   | 1940-present     | ERA5 (2025)            |
| $G_0$                | MERRA-2 reanalysis                | 1980-present     | GMAO (2025)            |

Table 1. The Belsk's instruments and their working periods.

3. Lines 11-12. I suggest moving the parentheses "(i.e. energy weighted...)" before "reaching the Earth's surface..."

**In the revised manuscript, these lines have been changed to:**

"However, homogenisation of the amount of biologically effective solar energy (i.e. energy weighted according to the sensitivity of the selected biological process to solar radiation) reaching the Earth's surface over long periods is challenging due to changes in measurement methods and instruments." L.10-13.

4. Lines 28 and 54-56. Clearness index is usually defined (e.g. Liu and Jordan, 1960) as the ratio between horizontal global irradiance and the extraterrestrial (top of the atmosphere) irradiance. Instead, the authors use the clear sky index: i.e. the ratio between actual horizontal global irradiance and that corresponding to clear sky conditions (which can be simulated). I suggest using the denomination 'clear sky index', instead of 'clearness index'.

Clear-sky index has been used in the revised manuscript instead of "clearness index" :

"....the clear sky index (CI) (i.e. a quotient of the all-sky global solar irradiance (G) at the surface and the corresponding synthetic clear-sky value ( $G_0$ ) to account for combined cloud/aerosol scattering effects on UVR)". L. 56-58

5. Line 91. "well-maintained Brewer". I suggest saying something about the Brewer maintenance and stability.

A reference to our previous paper of the Belsk's Brewer instrument explaining its maintenance and stability has been added.

"The details of the Brewer maintenance can be found in Czerwińska and Krzyścin (2024a)". L105-106.

6. Line 120. What's the meaning of 'pre-calibrated'?

The word "pre-calibrated" has been replaced by "roughly" (see l. 154) to better reflect the status of constants for instruments supplied by the biometers manufacturers. It is clear from our long experience with many instruments that such producer constants required considerable re-evaluation.

"Subsequent UVR measurements included SL501 A # 927 (1993–1994) and #2011 (1995–2013), which were only roughly calibrated by the instrument manufacturer prior to shipment" L.135-136

7. Line 147. 'radiance' Do the authors mean 'radiation'?

Yes. It should be "radiation".

"To validate the corrected UV observations at Belsk, the long-term variability of BE radiation was also obtained from the UVR reconstruction models (Section 2.3) using proxies (TCO3 and DCI) from the ground-based observations and reanalysis datasets". L.164-165

8. Lines 177-178. Add (eryt), (vitD3), (psor) in the Figure caption, as appears in the plot legend.

"eryt", "vitD3", and "psor" have been added to Fig.1 caption.

"Figure 1. Normalised action spectra for the specific biological effects: erythema appearance (eryt), photosynthesis of previtamin D3 in human skin (vitD3), psoriasis clearing (psor). "L. 197-198.

9. Line 199. Why the authors want to "allow for greater variability in the CC values"? I think the sentences in lines 514-519 do contribute to clarify this question. Thus, I suggest to move that explanation to Section 2.3.2.

In the submitted manuscript we had "...In order to allow for greater variability in the CC values, different criteria for clear sky conditions were applied, and the smoothing procedure was applied to the long (1976–2013) and short (1993–2013) UV time series for the CC1 and CC2 versions, respectively. "This statement has been rewritten following the reviewer's suggestion:

"Different criteria for the selection of cloudless days would result in even greater differences between the two CC versions. In addition, the smoothing procedure was applied to the long (1976-2013) and short (1993-2013) UVR time series for the CC1 and CC2 versions, respectively. We would like to have two different sets of correction coefficients to find out how the long-term pattern of biologically effective radiation is sensitive to the corrections." L. 234-237.

The sentence (l. 515-519 in previous manuscript) has been moved to Sect. 2.3.3.

"Model simulations of erythemal DRE and noon UVI under cloudless sky provide a basis for the correction procedure of the past UVR data. A selection of clear-sky conditions throughout the entire day from the daily proxy values (relative sunshine duration and DCI), which were available for Belsk, is not straightforward as only the examination of the daily course of these measurements would allow to capture cloudless moments within the day. Thus, two different sets of correction coefficients are proposed, called CC1 and CC2.". L.203-207

- 10. Line 231. Reference to Outer (2010) appears as 'den Outer' in the references list (and in line 58).
- "Outer" in line 58 has been replaced by "den Outer" in the revised manuscript (L.60).
- 11. *Lines* 247-248. *Include in the heading of Table 1 some reference to the Mod1 empirical model.*

The heading of Table 2 (Table 1 has been added) has been changed to:

"Table 2. Estimates of the regression coefficients,  $\alpha$  and  $\beta$ , describing the attenuation of erythemal DRE by the empirical model, Mod 1, defined by Eqs. (5–6), for the three SZA ranges at noon (SZAn). L. 285-286

12. Section 2.4. Why the authors use statistics over relative values  $(z_i)$  and not directly over the values? Could this be biasing the results by giving excessive weight to low values?

The relative values (in percent of the reference annual level from the observed data) are used after summing all the daily (Mod1) and monthly (Mod2 and Mod3) erythemal radiant exposures (in J/m2) over the whole year and summer for each year between 1976 and 2023. Therefore, the bias mentioned above is not present in this case.

13. I suggest, when assessing agreement between two series or sets of data, defininig the relative difference by subtracting the reference values in the numerator: this would change the sign of the differences, giving positive values when the tested value overestimates with respect the reference value.

Eq.(9) has been modified as suggested by the reviewer and the results in the following Tables 4-5 and A1 have different signs for the mean relative deviation.

14. Also, I suggest using the term 'deviation' instead of 'error'; this would lead to use MRD, MAD, SD and RMSD instead of MRE, MAE, SE and RMSE.

We agree and the reviewer suggestions has been applied (note new definitions by Eqs. (10–13 and MRD, MAD, SD, and RMSD in Tables)

15. *Line 339. The reference to LOWESS has been already given in line 195.*

This has been corrected and LOWESS definition is in line 219.

16. Line 609. Reference to Blumthaler et al. (1989) should be before those to Borkowski.

In the revised manuscript, Blumthaler et al. (1989) appears before Borkowski (1998).

---

## Author Comment (AC2)

**Reviewer 2**

General comments:

Reviewer 2 stated that the manuscript needed to be improved before publication, as indicated in the text below:

"*I endorse the publication and feel that the journal of Earth System Science Data is an appropriate place to present this important dataset. However, I also feel that the presentation needs to be improved. The manuscript is not an easy read, in particular Section 2.3.4. Readability and usefulness of the article could be greatly enhanced if the authors were to better describe the principles of the correction methods in one or two sentences at the start of each subsection before delving into details. I also find that the nomenclature is sometimes confusing as explained in more detail in my specific comments*."

To improve the manuscript, we have added explanations at the beginning of the following sections:

- Sect.2.3.1:

"Radiative transfer model simulations for clear sky conditions are used to quantify and correct biases in the output of the Belsk UVR radiometers. To speed up the calculations, the look-up tables were obtained using the Tropospheric Ultraviolet and Visible (TUV) Radiative Transfer Model (TUV, 2024)." L. 184-186.

- Sect.2.3.2

"Model simulations of erythemal DRE and noon UVI under cloudless sky provide a basis for the correction procedure of raw UVR data. A selection of clear-sky conditions throughout the entire day from the daily proxy values (relative sunshine duration and DCI), which were available for Belsk, is not straightforward as only the examination of the daily course of these measurements would allow to capture cloudless moments within the day. Thus, two very different sets of correction coefficients are proposed, called CC1 and CC2." L.203-207

- Sect.2.3.4

"The CCs  described in section 2.3.2 were obtained for cloudless conditions and applied to all-sky conditions, where the contribution of the diffuse part of the radiation increases with cloud cover and dominates under overcast conditions. It cannot be excluded that the instruments used to monitor UVR at Belsk have their own specific characteristics for recording diffuse radiation, and that $CF_{EFF}^*$ and $CF_{EFF}$ in Eqs. (3−4) should also depend on the combined characteristics of clouds and instruments. To test whether this is the case, we investigated how different regression models, which were trained using the UVR data collected between 2014 and 2023 (for this period, the quality of the broadband radiometer was confirmed by the Brewer Mark II observations), reproduce the daily doses of erythemal radiation throughout the 1976-2023 monitoring period.

The first model (Mod1) is based on clear-sky spectra determined with the RT model discussed in Section 2.3.1 and a cloud modification factor (CMF) derived from DCI data. The second and third models (Mod2 and Mod3) are based on $TCO_3$ and DCI data evaluated on a monthly basis. $TCO_3$ and DCI were either taken from observations at Belsk (Mod2) or ERA5 reanalysis (Mod3)." L. 257-268.

We have done our best to improve the readability and usefulness of the manuscript by responding to the reviewer's specific and technical comments

**Specific comments**

"*L14, L37: I believe the term "biometer" is only used by Solar Light and Yankee Environmental Systems but no other companies. Since the authors also present data from the original*

*Robertson-Berger meter and the Kipp and Zonen UV radiometer, it would be good to use a more neutral term such as "broadband radiometer with erythemal response".*

In the revised text we use the "more neutral term" - erythemal broadband meter (or radiometer). These term has been widely used in the literature when discussing different types of instruments that measure erythema-weighted UV radiation. For example, see the titles of the papers by Hülsen, and Gröbner (2007) Gröbner et al., 2009, Schmalwieser et al., 2022) that appear in the references. In the text, "biometer" appears only when the Solar Light instruments with the commercial name containing the word "biometer" are discussed.

*L25-28: This should be better explained. First state that two different homogenization methods were applied to the measurements of the various instruments to derive a consistent dataset for the period 1976–2023. Then state that this dataset was further validated by reconstructing independent datasets of DRE and the UV index from proxy data (i.e., total column ozone, aerosol optical depth, and the global irradiance clearness index). When I first read the article it was not immediately clear to me what the "three regression models" entailed.*

These sentences have been changed. In the revised manuscript, we explain that:

"Three regression models of the erythemal data on common UVR indices (total column ozone, aerosol optical depth and global clear sky irradiance index) were used to reconstruct the UVR data from the beginning of the Belsk observations, allowing further validation of the homogenised UVR data" . L.27-30

*L68: Please provide a reference for the "Dave-Halpern model". (It is not clear whether the reference Słomka and Słomka (1985) also includes a description of this model.).*

Reference to Dave-Halpern model (Dave and Halpern, 1976) has been added (L.69-70)

*L88 and L188: The UV index is not the midday value of erythemal irradiance. It is erythemal irradiance multiplied with 40 m2/W.*

These two lines have been corrected as UV index is a dimensionless measure of the erythemal irradiance. In the revised manuscript, we have:

"The re-evaluation for the period 1976−2013 is based on a comparison of the measurements with the synthetic daily erythemal radiant exposure and UV index at noon from a radiative model simulation". L.101-103

"For the first period, only the erythemal DRE were archived, whereas for other periods daily maximum of UV index (UVI$_{MAX}$) was also available (equal to the value of erythemal irradiance at noon during a cloudless day)." L. 211-213

*L103: Why "May 1975"? I thought measurements started in 1976 according to the paper's title?*

The first observation by the erythemal broadband radiometer was in May 1975, and the regular monitoring started on 1$^{st}$ of January 1976. This was explained in the revised text:

"Recording of erythema irradiance with a standard RB meter (detector recorder No. 40) started in May 1975 at Belsk, but continuous monitoring began on 1 January 1976 and lasted until 1994" L.117-118

*L138: A symbol for the "daily CI" should be introduced, such as CI_d. I got confused later in the manuscript because I recalled from line 54 that CI is an instantaneous measurement while later in the manuscript all modelling is based on the daily CI.*

In the revised manuscript, DCI abbreviation appears when discussing the daily values of clear-sky index. DCI was defined as follows:

"The daily representative of CI, DCI, which is further used in regression models (Sect. 2.3.4), is calculated as the quotient of the daily integrals (sunrise to sunset) of G and G$_0$." L. 156-157

*L147: "To support the quality of the UV observations at Belsk," > "To validate the corrected UV observations at Belsk"*

OK. The statement has been changes according use the reviewer's suggestion. L.164

*L166: To improve readability, start this section with describing what the radiative transfer model is used for. For example, state that the TUV radiative transfer model is used to quantify and correct biases in the UV radiation measurements of the radiometers.*

We added the following lines, taking into account the reviewer's suggestion.

"Radiative transfer model simulations for clear sky conditions are used to quantify and correct biases in the output of the Belsk UVR radiometers. To speed up the calculations, the look-up tables were obtained using the Tropospheric Ultraviolet and Visible (TUV) Radiative Transfer Model (TUV, 2024)." L. 184-186.

L179: Why are you using the monthly mean AOD? Why not higher frequency? Why not use daily AOD on days where AOD is available?

AOD measurements were infrequent (a few per month), especially before the start of automatic AERONET measurements in 2004. We use monthly averages to account for the atmospheric cleaning trend resulting from the government's policy to reduce atmospheric pollution from anthropogenic dust.

*L180-181: "$RE\_EFF,Clear-Sky$" > "$RE\_EFF,CS$". ("CS was used earlier to indicate clear sky, please use consistent acronyms of variables throughout the manuscript).*

OK. It has been corrected according to the reviewer's comment.

*L193: Regarding "for the days when clear sky conditions can be assumed from the ancillary data.": Do you mean days that were clear sky from sunrise to sunset? If not, please clarify.*

Yes, the clear sky day is from sunrise to sunset. This statement has been changed to make it clear:

"…for days when ancillary data indicated that the sky was clear throughout the day" L. 217-218.

*L192-198: The terms "multiplier" and "calibration coefficients" are not clear and this caused confusion when I first read the manuscript. What you define as "raw erythemal data" on line 192 are not really raw data (which would be volts or amps). They are calibrated data (e.g., erythemal exposure or the UV index) affected by drifts and other artefacts. So what you refer to as "multipliers" or "calibration coefficients" are actually correction factors that are applied to calibrated data. I may sound picky pointing this out, but by using more appropriate terms, the readability of the manuscript could be greatly improved. The term "calibration procedure" that you use on line 192 should also be changed to "correction procedure". To be consistent with this new nomenclature, CC1 and CC2 should also be renamed to CF1 and CF2 for correction factor 1 and 2. (Although this suggestion would be in conflict with using CF for "conversion factor" in Section 2.3.3. So please be creative and find nomenclature that better describes the various factors than that are used in the manuscript.) Also, what does "locally weighted" mean on line 194?*

According to the reviewer's comments, in the revised manuscript, the "calibration coefficients" have been changed to the "correction coefficients" and consequently "calibration procedure" has been replaced by "correction procedure". In addition, "Locally weighted" is the name of the smoother introduced by Cleveland (1979). To make it clear, the previous sentence (l. 194)) has been replaced:

"LOcally Weighted Scatterplot Smoothing (LOWESS) proposed by Cleveland (1979) was used to extract the smoothed pattern of the multipliers…" L.218-220

*L199: Why would you "allow for greater variability in the CC values". Ideally, the CC values are the best estimate of the correction for a given day. Why would you like to have "greater variability" for this correction?*

We would like to have two different sets of the correction coefficients to find out how the results (long-term pattern of biologically effective radiation) are sensitive to such corrections. The statement " allow for greater variability in the CC values" has been deleted and replaced by the following one:

"Different criteria for the selection of cloudless days would result in even greater differences between the two CC versions. In addition, the smoothing procedure was applied to the long (1976-2013) and short (1993-2013) UVR time series for the CC1 and CC2 versions, respectively. We would like to have two different sets of correction coefficients to find out how the long-term pattern of biologically effective radiation is sensitive to the corrections." L. 234-237.

*L201: "Accordingly, the following conditions were applied for the selection of clear sky sets:" > "The following conditions were applied for the selection of clear sky data used in the two correction methods:"*

The statement proposed by the reviews has been used in the revised manuscript. L.221-222

*L207-208: CC2 is based on UVI at noon. If so, why does this depend on sunshine hours for SZA < 85. Whether or not the Sun is shining at times other than noon is irrelevant for the noontime UVI. (If I understand correctly, you only need to ensure that there is clear sky at noon when the comparison between the measurements and model takes place).*

Yes, reviewer 2 got it right. The information on all-day clear-sky conditions indicated that this maximum occurred at midday. This means that we can model the irradiance at solar noon.

*L209-210: I don't understand this sentence. How can a recalibration in 2011 be informative for a period prior to 1 January 1993? Do you mean that data collected prior to 1993 were assessed in 2011?*

The original RB data (1976-1992) were fitted to the reconstructed DRE with a regression model using $TCO_3$ and $G_o$ as explanatory variables (see Eq. (5) and Figs. (5-6) in Krzyścin et al. (2011)). Figure 6 (this manuscript) confirms that the calibration of the RB data in 2011 was correct, so the current CC2 values can be set to 1 for the entire RB measurement period. The following statement has been added to explain this approach:

"Prior to this date, a re-evaluation of the RB data with a model mimicking the KZ radiometer measurements by Krzyścin et al. (2011) showed that the correction was not necessary, i.e. CC2=1. This choice is also confirmed here by the constant long-term patterns of CC1 in the period 1976-1992 (Fig. 6a), and only a small jump in the differences between CC1 and CC2 in 1992/1993 (Fig. 6b)" L. 230-233.

*L220-225: "current D day" > "day D". (If I understand correctly, D indicates a specific day within the period 1996-2023. So it is not the "current" day.) Also the meaning of the asterisk in D\* is a bit murky. You may say that the course of the SZA on a given day of the year is more or less the same in every year. Hence, the conversion factor only considers the TCO for the day in question plus the course of the SZA for the day of the year that corresponds to that day.*

Yes, we assume that the course of the SZA on a given day of the year is more or less the same in every year, so the multipliers of the erythemal daily radiant exposure depend on actual TCO3 and Julian day of the year (denotes as JD in the revised manuscript). The avoid misunderstanding of previous notation (D and D\*) we rewrite the text using notation YY, MM, DD for the calendar day and JD for the corresponding Julian Day:

"Following this concept, the daily radiant exposure for previtamin $D_3$ synthesis and psoriasis clearance in year (YR), month (MM), and day of month (DD) $RE_{VITD3}(YR, MM, DD)$ and $RE_{PSOR}(YR, MM, DD)$, were estimated using the daily conversion factor, $CF^*_{EFF}$, applied to the erythemal DRE ( $RE_{ERYT}(YR, MM, DD)$ ):

$$RE_{EFF}(YR, MM, DD) = CF^*_{EFF}(TCO_3, JD) \times RE_{ERYT}(YR, MM, DD), \ EFF = \{VITD3, PSOR\}, \quad (4)$$

where $CF^*_{EFF}$ depends on $TCO_3$ and $JD$ (Julian day number corresponding to the current day YR:MM:DD). $CF^*_{EFF}$ and $CF_{EFF}$ values were obtained from radiative transfer model simulations. The time series (1976−2023) of the conversion factors, $RE_{EFF}(YR, MM, DD)$, and the corresponding noon value of the biologically effective irradiance $Ir_{EFF}(t = \text{noon})$ have been archived in the IG PAS Data Portal (Krzyścin, 2024)." L. 247-255.

*L229-230: This sentence should be improved and extended. Please better describe what you did, e.g.: "We developed several regression models from data of the period 2014−2023 by correlating measured radiant UV exposures against UV exposures calculated from proxy data*

*such as TCO and the daily clearness index. We then applied these regression models to proxy data of the entire period (1976-2023) to provide a quality measure of the corrected UV datasets." Since three regression model are considered, these should also be briefly introduced here so that the reader knows what to expect in the remainder of this section. For example: "The first model (Mod1) is based on clear-sky spectra determined with the RT model discussed in Section 2.3.1 and a cloud modification factor derived from CI data. The second and third model (Mod2 and Mod2) are based on TCO and short-wave irradiance (G) data evaluated on a monthly basis. TCO and short-wave irradiance were either taken from observations at Belsk (Mod2) or ERA5 reanalysis (Mod3)."*

The mentioned sentences have been replaced by new text following the reviewer's comments:

"The CCs  described in section 2.3.2 were obtained for cloudless conditions and applied to all-sky conditions, where the contribution of the diffuse part of the radiation increases with cloud cover and dominates under overcast conditions. It cannot be excluded that the instruments used to monitor UVR at Belsk have their own specific characteristics for recording diffuse radiation, and that $CF_{EFF}^*$ and $CF_{EFF}$ in Eqs. (3−4) should also depend on the combined characteristics of clouds and instruments. To test whether this is the case, we investigated how different regression models, which were trained using the UVR data collected between 2014 and 2023 (for this period, the quality of the broadband radiometer was confirmed by the Brewer Mark II observations), reproduce the daily doses of erythemal radiation throughout the 1976-2023 monitoring period.

The first model (Mod1) is based on clear-sky spectra determined with the RT model discussed in Section 2.3.1 and a cloud modification factor (CMF) derived from DCI data. The second and third models (Mod2 and Mod3) are based on $TCO_3$ and DCI data evaluated on a monthly basis. $TCO_3$ and DCI were either taken from observations at Belsk (Mod2) or ERA5 reanalysis (Mod3)." L. 257-268

*L233: The CI was defined on line 55 as the "quotient of the all-sky global solar irradiance (GSI) at the surface and the corresponding synthetic clear-sky value to account for combined cloud/aerosol scattering effects." Hence, CI is a function of the instantaneous irradiance not a daily value, as CI(D) implies. So what is CI(D)? is it a daily average? Clear definitions of CI, CI(D) and CMF should be provided. (If CI(D) is the daily average, CI(D) is not a good acronym as such a notation would mean "instantaneous clearness index as a function of day D" with D being the argument of CI. CI_D would be a better variable name. (Underscore indicates a subscript.))*

CI was mentioned in Introduction as a possible proxy to model the biologically effective irradiance. As our models estimate daily erythemal radiant exposures for all-sky conditions, the daily clear-sky index (DCI) was applied further in reconstructions of the biologically effective radiant exposures. DCI was defined as:

"The daily representative of CI, DCI, which is further used in regression models (Sect. 2.3.4), is calculated as the quotient of the daily integrals (sunrise to sunset) of G and $G_0$" L. 156-157

*L301: Figure 3a does not show a ratio, as the caption states, but is a correlation or scatter plot.*

This statement has been rewritten. Taking into account the reviewer's comment, we have:

" An example of such a monthly comparison (for June 2023) is shown in the scatter plot between the BS064 and KZ616 erythemal irradiances measured under clear-sky conditions (Fig.3a). In addition, Fig. 3b shows the monthly ratios between these clear sky erythemal irradiances for the entire BS and KZ comparison period (2014-2023).". L. 334-337.

*L330: Again the use of the term "calibration coefficients" is confusing in this context. Please use "correction factors" or something similar. (Thank you for using the term "correction method" on line 337 and not "calibration method"!)*

In the revised manuscripts, when discussing the raw data multipliers, "calibration" has been replaced by "correction".

*L336: "are proposed (Sect. 2.3.2) using" > "were proposed in Sect. 2.3.2 using"*

Done. L.376

*L341: Please specify range of the "former" period.*

*"Former"* has been replaced by 1976-1992

"In the 1976-1992 period, UVI values were not archived." L.381

*L355-356: Describe better and use different colors as those used in panel (a). What's shown are not differences between calibration coefficients but the difference "(Modelled RE - Observed RE) minus (Modelled UVI - Observed UVI)".*

Different colours have been used in Fig.6a and Fig.6b (brown and black instead of blue and red). Term "difference" has been used (L.395).

*L401: Regarding: "to fill gaps in the proxy data": Do you mean fill gaps in the measurements of the biometers? (I thought the ERA5 reanalysis data _are_ the proxy data.*

ERA5 provides proxies (TCO3 and DGI) used in Mod3 to reconstruct monthly erythemal radiant exposures. Thus, the agreement between MOD2 (proxy from the measurements at Belsk) MOD3 means also possibility to simulate UVR for a site without ground-based measurements of the proxy values. The statement regarding 'gap-filling' has been removed as such 'filling' was not necessary.

*L518: I would not conclude that the CC1 and CC2 methods "very different." In fact, they are quite similar. Just one is based on daily exposure while the other one on the noontime UVI. If the responsivity of a biometer is off by a given factor, it would affect daily and noontime values equally. It is therefore not surprising that CC1 and CC2 results are similar.*

In fact, CC1 and CC2 are different as the former coefficients characterise the daily course of UV radiation but the latter ones are related to the maximum of erythemal irradiance usually near the noon. The UVR forcing variables, TCO3 and AOD, can differ throughout the day as these variables sometimes have large intra-day course, affecting values of the former coefficients.

*L548: Did the instrument used by Chubarova also have an erythemal response or did it have a constant response over the 300-380 nm range? If the latter, the signal would be dominated by wavelengths in the UV-A; hence, the sensitivity to variations in ozone would be minimal.*

The instrument used in Moscow was a broadband instrument and its output was mainly due to the UV-A part of UVR, which is weakly affected by atmospheric ozone. It cannot be used directly to quantify the risk of many biological processes that are sensitive to wavelengths in the UV-B range (290-315 nm) but can be used to parameterise cloud effects on UV-B in a reconstruction of the erythemal radiation.

*L560-568: I would think that Sections 6 should come before Section 5.*

According to submission guidelines of the journal, data availability should come before conclusions.

 **Technical comments**

*"I presume the article will be prove-read by a copy editor. I therefore keep my comments regarding language to a minimum."*

All reviewer's comments have been incorporated into the revised text. They are as follows:

*Always add "radiation" after "UV" or define "UVR" as "ultraviolet radiation" (e.g., in line 52, change "surface UV modelling" to "modelling of surface UV radiation") This comment applies to many instances.*

'UVR' is defined (L. 43) and is used throughout the text.

*L2, L166, 177: Change "erythema appearance" to just "erythema". The term "erythema appearance" is not commonly used.*

The term "erythema appearance" has been replaced by "erythema".

*L3: "from erythema biometers" > "derived from broadband radiometers with erythemal response" (The term erythema biometers does not describe the instrument and is not widely used.).*

In the revised text, "from erythema biometers" has been deleted and replaced by "derived from erythemal broadband meter". L.3

*L17: "the appearance of erythema," > "erythema"*

The term "erythema appearance" has been replaced by "erythema".L.17

*L40: penetrate > reach*

Done. L.42

*L41: "were destroyed" > "would be destroyed"*

Done. L.43

*L73. Please explain acronym "IG PAS" (presumably Institute of Geophysics of the Polish Academy of Sciences, but this is not mentioned).*

This acronym was explained earlier in the text. L.68

*L76: Mod > Model.*

Done. L.78

*L107: "MED=210 J_eryt m-2," > "1 MED=210 J_eryt m-2,"*

Done. L.122

*L126: proved > proven*

Done. L.142

L137: "CI is a commonly used" > "The CI is a commonly used"

This expression has been replaced by:

"The daily representative of CI, DCI, which is further used in regression models (Sect. 2.3.4), is calculated as the quotient of the daily integrals (sunrise to sunset) of G and $G_0$". L.156-157.

CI has been defined earlier:

"the clear sky index (CI) (i.e. a quotient of the all-sky global solar irradiance (G) at the surface and the corresponding synthetic clear-sky value ($G_0$) to account for combined cloud/aerosol scattering effects on UVR)". L.56-58

*L167: Move "irradiance at noon in day D" before "Ir_EFF"*

Done. L.188

*L172: "Ir($\lambda$, t)" should be "Ir_CS($\lambda$, t)". Change "in time t for the wavelength" to "at time t and at wavelength"*

Done. L.192

*L192: "The calibration procedure" > "The correction procedure*

Done. L.203-204

*L204: "difference between observed sunshine duration and theoretical one" > "difference between the observed sunshine duration and the theoretical one"*

This has been changes to:

"the daily difference between the observed and the theoretical maximum sunshine …". L.224

*L205: "0.5 hour as for higher" > "30 minutes. This limit was chosen because broadband UV measurements at larger SZAs …"*

Done. L. 225

*L232: "that the erythemal" > "the erythemal"*

Done. L.270.

*L253: "in the UV explaining variables X" > "in variables X that affect UV radiation"*

Done. L.292

*L281: Opening parenthesis missing in the formula for F.*

Done. L.321

*L283: "(for Rk+1 <0, F=1):" > "If Rk+1 is smaller than 1, F is set to 1."*

This has been replaced by the statement:

"F is set to 1 if the autocorrelation coefficient in the residual time series is negative." L.323-324

*L304: (CC ver. 1) and What does "(CC ver. 1) mean? This was not defined earlier. Is this equal to CC1? Please use consistent nomenclature!*

CC ver.1 appeared only here. This has been replaced by CC1. L.346

*L307: What do you mean with "CF values" here? CF values were earlier defined as conversion factors from erythema to other weighting functions. I suspect you mean CC1 values here. Again, please use consistent nomenclature!*

"CC1" has replaced misprinted "CF". L.349

*L311: "as shown by the linear regressions close to the 1-1 perfect agreement line in the three scatter plots (Fig. 5)." > as shown in the scatter plots of Fig. 5, which indicate that Brewer and biometer data cluster about the ideal 1:1 line.*

Done. L.353-354

*L322: "used in routine" > "measurements used in routine"*

Done. L.362

*L369: see > Note*

Done. L.413

*L427: "determination coefficients" > "coefficients of determination"*

Done. L.463

*L436: thesis > hypothesis*

Done. L.472

*L497 comes > are*

Done. L.90

*L521: "as shown by the comparisons with BS64 data" > "because they agreed well with the BS64 data"*

Done. L.539

*L598-599: the title "References" appears twice.*

The second "Reference" has been deleted.

---

## Author Response (AR2)

**Response to the Reviewer Comments**
**(the reviewer's text is in italic)**

**General comments:**

*" I feel that the manuscript is now almost ready for publication. However, there are still a few issues (discussed below) that should be taken into consideration. Most importantly, the uncertainty estimates should be changed from the 1-sigma level to the 2-sigma level to be more in line with similar estimates of trends reported for other locations (I apologize that I didn't catch this in my 1st review). Furthermore, some parts of the manuscript could still be improved to better readability."*

In the revised manuscript, the standard errors in Tables (6, 7 and 8) have been changed to the estimates corresponding with the 95% confidence level. Namely the previous standard errors are multiplied by 2.05 and 2.10 for the 1976-2004 (29 yr.) and 2005-2023 (19 yr.), respectively,

"Bold font indicates a statistically significant trend value at the 95% confidence level, based on the standard error of the trend multiplied by the corresponding critical T-value ($T_c$) for a two-sided probability. For the periods 1976–2004 and 2005–2023, $T_c$ is 2.05 and 2.10, respectively, with 28 and 18 degrees of freedom "L. 494-496

The parts of manuscript required correction, which are mentioned below, have been taken into account, as stated further in this text.

Specific major comments:

*" .... To be more in line with many publications that have estimate trends in UV radiation and their uncertainty at other locations, I suggest that the authors use the same approach and report the uncertainties of their trend estimates for a confidence level of 95%. For a large number of samples (which is the case here), the Student's t statistic is approximately two. Hence the uncertainty values provided in Tables 6 and 7 could be multiplied with 2 to refer to a confidence level of 95% (although technically, it would be a confidence level of 95.45% as 95.45% of values of a normal distribution are within +/- 2 sigma). When using this approach, the number printed in bold in Tables 6 and 7 (indicating significant trends at the 2-sigma level) would also align with the actual numbers of the uncertainty estimate."*

According to the reviewer's suggestion, the uncertainties of the trend estimates are shown in Tables 6-8 with the 95 % confidence interval, i.e. the standard error of the slope has been multiplied by factor 2.05 (degree of freedom =28, 1976-2004) or 2.05 (degree of freedom=29, 2005-2023) to obtain the trend error at this confidence level.

*L3811ff: I think the assumption that CC2 values are equal to 1 over the period 1976−1992 cannot be supported with the comparison of simultaneous measurements for the periods 1992−1994. Results from that comparison cannot rule out that the sensitivity of the instrument has changed between 1976 and 1992. Fortunately, this does not seem to be the case because the ratios of the modelled and observed DRE values shown in Figure 6a (blue dataset) is rather flat. Hence, the wording in the paragraph starting on line 381 should be toned down, stating that a constant value of 1 is just an assumption that is not supported by data and that the conclusion that the instrument has not drifted significantly over this period can only be established from the CC1 values. This also means that the "oscillation with 0.015 amplitude" mentioned on line 388 is not supported with real data and should be removed.*

We agree with this and explain it in the revised manuscript:

'.....However, CC2 values were assumed to be equal to 1 because the RB meter was previously adjusted to the SL501A #927 output using simultaneous measurements taken during the 1992–1994 period (Puchalski et al., 1995). Prior to 1992, CC2 =1 could be inferred from a flat CC1 pattern based on the daily erythemal RE." L. 384-386.

In addition, we remove the text 'oscillation with 0.015 amplitude' and consequently we delete also Fig.6b.

*L433–436: I don't understand the sentence starting with "The performance". Please rephrase and better explain the rationale.*
This part has been changed. We explain in the revised text:
'In most cases, Mod2 and Mod3 outperform Mod1 (see Table 5), with smaller values of descriptive statistics. This is because these models are based on relative monthly differences from the respective long-term means for the periods 1976–1992, 1993–2013 and 2014–2023. These means were obtained by averaging the re-evaluated RB meter and SL501A, and the original KZ616 data. The Mod1 model did not apply a constraint on the average UVR in these sub-periods." L.428-432.

*L586: The Conclusions are rather disconnected from the rest of the paper and some conclusions are not supported by the paper's results. For example, overlapping datasets were not discussed sufficiently in this paper. So this statement is not supported by the results and should therefore be removed. Please consider improving the Conclusions.*
Conclusions have been reworded and the term 'overlapping data sets' has been precisely defined to strengthen the link to the results obtained in the manuscript. As this paper falls into the 'data paper' category, in order to support the suitability of the data for future analysis, a deeper analysis of the data will be published in other journals, and these activities are ongoing.
"It is generally accepted that the use of a sample of time series containing different possible realisations of a time series increases the reliability of the results compared to the analysis of a single time series. Therefore, this study includes 7 time series (refer to number of analysed time series shown in Table 7) to discuss the reliability of year-to-year variability and trends in annual and summer biologically effective radiant exposures. This is beneficial for evaluating the quality of the data and establishing confidence in the results. Data archived in PANGEA (Krzyścin et al., 2024) and in the IG PAN Data Portal (Krzyścin, 2024), together with the results of three regression models, form the reliable basis for analysing UVR time series at Belsk for the period 1976–2023. The long-term variability of erythemal radiation calculated for Belsk corresponds to that previously recorded at distant stations in central/eastern Europe, making results of these future analyses applicable to wider areas. For example, the daily characteristics of BE radiation at Belsk seem to allow the development of scenarios for outdoor human activities, enabling people to obtain the maximum health benefits from sunbathing while minimising the risk of erythemal overexposure. L.585-596

**Specific minor comments:**

*L120-121: It is described here that sunburn units are converted to MED. However, in the remainder of the manuscript either the UV dose or the UV index are discussed. Hence, it would be good to mention explicitly that MEDs were converted to the erythemal dose by multiplication with 210 J m-2.*
We agree with this and explain it in the revised manuscript:
"…MEDs were converted to the erythemal doses by multiplication with 210 $J_{eryt}$ m$^{-2...}$"L. 122-123.

L136: *"roughly calibrated" sounds awkward. I would be better to say that these instruments were calibrated by the manufacturer but that these calibrations are subject to large uncertainties. Then emphasize again that one objective of the work presented in this manuscript is to adjust the manufacturer's calibration in order to reduce these uncertainties.*
We agree with this and explain it in the revised manuscript:
"Subsequent UVR measurements included SL501A # 927 (1993−1994) and #2011 (1995−2013), which were calibrated by the instrument manufacturer prior to shipment, but these calibrations proved to be very inaccurate Therefore, this paper is another attempt to recalculate past UVR data" L. 135-137.

*L148: "daily average TCO3 measurements throughout the day" is awkward. A daily average is one number per day. Just delete "throughout the day".*
We agree and "throughout the day" has been deleted. L. 149.

*L180: Please state the "typical values" that were used, in particular the values of the single scattering albedo and the asymmetry parameter.*
We agree with this and explain it in the revised manuscript:
" Taking into account climatology of the Belsk's aerosol characteristics (AERONET, 2025) values of 0.95 and 0.69 are taken for single scattering albedo and asymmetry factor, respectively." L.181-183.

*L186: In addition to TUV (2025), please also consider to cite the paper introducing the TUV model: "Madronich, S.: UV radiation in the natural and perturbed atmosphere, in: UV-B Radiation and Ozone depletion. Effects on Humans, Animals, Plants, Microorganisms, and Materials, edited by: Tevini, M., 2, Lewis Publishers, Boca Raton, 17-69, 1993."*
Reference to Madronich (1993) has been included in the revised manuscript.
"The UV model, which was introduced by Madronich (1993), has since been widely used in UVR simulations."
L. 189-190.

*L212: Regarding "(equal to the value of erythemal irradiance at noon during a cloudless day)." If there were scattered clouds, which can increase the UVR beyond the noon clear sky value, was the noon value archived or the maximum UVI under those clouds?*
Daily maxima were archived from the measurements. Model simulations (TUV) of UVI under cloudless conditions give UVI maxima at local solar noon, so for cloudless conditions the two maxima are equal.

*L221: Adding to my previous comment: If UVI_max and not the noon values was archived, there would be a bias because the model only calculates the clear sky value at noon. Was this taken into consideration?*
YES. In the manuscript, the measured UVI maxima were compared with the modelled values at local solar noon only for clear sky conditions, which are inferred from the cloud proxy.

*L217: The term "synthetic" can be misleading. I would prefer using the word "modelled" throughout the document when referring to the results obtained with the radiative transfer model.*
We follow this recommendation and replace the word 'synthetic' with 'modelled' throughout the document.

*L234: it was not discussed previously what the difference between CC1 and CC2 results is. So mentioning "even greater differences" is odd if there is no reference.*
We agree: the line containing "even greater differences" has been deleted.

*L248 and similar: I find it odd to use symbols with two letters for year, month and day of the month. Why not just use Y, M, and D?*
We agree and Y, M, and D have been used in the revised manuscript.

*L252: Do you really mean "Julian Day" here? The Julian Day is the day since 1 January 4713 BC (e.g., https://en.wikipedia.org/wiki/Julian_day). Is that what you mean or do you mean the "day of the year" (a value between 1 and 366)?*
We agree and use "day of the year" in Eq. (4) and in the text (L.254).

*L336: regarding "monthly ratios": Do you mean: "monthly averages of the ratios of BS064 and KZ616 measurements during clear skies"? If so, please say so!*
We agree and explain:
"In addition, Figure 3b shows the monthly mean ratios between BS64 and KZ616 erythemal irradiance obtained during cloud-free periods from 2014 to 2023." L.338-339.

*Paragraph starting on line 348: It is confusing that the discussion jumps between the Brewer/KZ comparison in Figures 2 and 3 to the model comparison in Figure 4 and then*

*back to the Brewer–KZ comparison in Figure 5. I suggest to move Figure 5 before Figure 4, and discuss the annual cycle of the correction coefficient CC1 at the end.*
It has been done according the reviewer suggestion. Fig.5 has been moved before Fig.4 and the discussion on the CC1 pattern has appeared at the end.

*L454: Mod1 does not use "re-evaluated ERE values". It uses radiative transfer calculations and CMFs.*
We agree with this and explain it in the revised manuscript:
"In the case of Mod1, the erythemal annual and summer RE are constructed using the results of the TUV model and the CMFs estimated from the DCI values" L. 450-452.

*L472: Since TCO3 and cloud transparency data are available, it should be analyzed whether these large fluctuations are caused by either or both of the two factors.*
We agree with this and explain it in the revised manuscript:
"As $TCO_3$ and cloud transparency data are available, it is important to analyse whether these large fluctuations are due to one or both of these two factors." L. 469-470.

*L545: "standard error of ~1 % per decade" should be changed to "standard error of ~2 % per decade at a 95% confidence levels" per my "major" comment above.*

The mentioned standard error of ~1 % per decade was derived averaging the trend values shown in Table 7 for various models (in total 7 models participated in the calculations), which are shown in the column for the period 1976-2004. This is not the standard error of the slope of the regression. In Table 7, for each model we calculate the 95% confidence range (slightly larger than 2xstandard slope error). Further in the text we explain:
"The uncertainty (at the 95% confidence level) of the individual trend estimates for the period 1976-2005 (Table 7) are of about 2-3% per decade, i.e. quite close to double the standard error of the mean trend derived from the sample." L. 546-548.

**Technical comments**

*L17: "the erythema" > erythema*
Done (L. 17)
*L20: Robertson-Berger > Robertson-Berger meter, Kipp-Zonen > Kipp & Zonen (please use consistent spelling with "&" throughout!*
Made using the WORD option to replace one word with another.

*L25: Brewer spectrometer > Brewer spectrophotometer (at least that's the official name of the instrument, although it is not really a photometer*
Done (L. 25)

*L28: "on common UVR indices" Don't use "indices" since this can be confused with the UV Index. Use either "proxy data" or rephrase, e.g.: "depending on the most important factors affecting UVR (total column ozone …)"*
We use reviewer recommended "proxy data" (L. 28).

*L44: "in the catalytic" > "in a catalytic" or " in ... cycles*
Done (L.44)

*L59: effective > "important factors".*
Done (L.59).

*L69: "the radiative" > "a radiative"*
Done (L.69).

*L71: "were their results in" > "were that their results were reported in".*
Done (L.71-72).

*L78: I note that the Solar Light Biometer's model number is referred to as "501A" here, without a space between "501" and "A". In the rest of the manuscript, the model name is referred to as "501 A", i.e., with a space between "501" and "A". I believe the spelling without a space is the correct one and should be used throughout the document. This would also improve readability since an orphaned "A" looks awkward.*

Throughout the text, in line with the reviewer's comments, '501A' has been used instead of '501 A'

*L94: "Database Network for the Detection of Atmospheric Composition Change (NDACC) include also stations" > The data archive of Network for the Detection of Atmospheric Composition Change (NDACC) also includes data of stations with at least three decades of".*

Done (L. 94-97).

*L102: "and UV index" > "and the UV index".*

Done (L. 102).

*L111: "calculations in" > calculations for"*

Done (L. 111).

*L112: "and versions of the recalculated" > "and three versions of recalculated"*

Done (L. 113).

*L126: "those with" > "those obtained with"*

Done (L. 126).

L142: involved > used

Done (L. 143).

*L167: What are "intra-day TCO3 values"? Do you mean "several TCO3 values throughout the day"?*

Exactly. Sometimes, it was around 40 measurements a day.

*Table 1: The Kipp & Zonen (not Kipp-Zonen) UV-AE-T serial number is given here as "30616". In the rest of the manuscript, the serial number of the instrument is referenced as "616". Please use the correct serial number throughout the document. Regarding the TCO3 row in the table, in addition to the Dobson measurements, also the source of the TCO3 satellite data used in the manuscript should be mentioned here.*

The abbreviation 'KZ616' (Kipp & Zonen UV-AE-T #30616) is defined in line 100 and used throughout the main text. The full name is used in Table 1. Satellite sources (from OMI and OMPS instruments) have been added to Table 1.

*L185: "the look-up" > look-up*

Done (L. 188).

*L190-191: Reverse the sequence of the two equations. Since Eq. (1) uses the result of Eq. (2) as input, Eq. (2) should be first.*

Done (L. 194-195).

*L197: The text uses upper case acronyms for eryt, vitD3 and psor. Please use consistent spelling.*

Done (see the new legend Fig.1 and L. 200-201).

*L120: "(t=noon)" > "for noon". Also please specify whether you refer to local solar noon or a specific time (e.g., 12:00) that remains constant throughout the year.*

It has been explained in the revised manuscript:

"… and the corresponding value of the biologically effective irradiance at local solar noon, $Ir_{EFF,MAX}$, have been archived in the IG PAS Data Portal…" L256-257.

*L208: Delete "intraday"*
Done (L. 211).

*L225: below > "less than"*
Done (L. 228).

*L259: excluded > "ruled out"*
Done (L. 261).

*L260: delete "should" or better, repeat "that"*
Done (L. 262).

L270: "the current day" > "a given day"
Done (L. 272).

*Eq. (7): The subscript in "c_k" should be upper case to be consistent with the spelling elsewhere*
Done (L. 296).

*L309: "model value" > "the value of the regression model" (to make clear that this is not the RT model)*
Done (L. 310).

*L331: "KZ616 replaced the raw SL501 A #2011," > "data of KZ616 replaced the raw data of SL510A"*
Done (L. 333).

*Figures 2 and 5: Replace "Biometer" with "EBR" on the title of the y-axis.*
Done (see new Fig.2 and Fig.4).

*L344: monthly > monthly average*
Done (L. 345).

*Figure 4: The title of the y-axis should be "Correction coefficient*
Done (see the y-axis of Fig.5).

*L398: Why "original"? There is only one KZ data set as these data were not corrected to my understanding.*
Yes. KZ616 data were not corrected, so we delete "orginal'(L.347, 355, 409, 432, and 515)

*L440: "The lowest correlation" > "The smallest correlation"*
Done (L. 436).

*L457: The nomenclature using parentheses to indicate summer data is confusing. Please split in two sentences if necessary.*

The splitting is:
"Fig.8a and Fig.8b are for the erythemal annual and summer RE with the use of CC1 but correspondingly Fig.A1a and Fig.A1b are based on CC2 values." L.455-456.

*L536: "to assess the uncertainty range of the correction method applied to the raw UVR data." > "to correct and homogenise the time series of erythemal data"*
Done (L. 534-535).

L555: "of about" > "of only about"
Done (L. 551)